# Sharp Inequalities between Total Variation and Hellinger Distances for Gaussian Mixtures

**Joonhyuk Jung** [1]   **Chao Gao** [1]

## Abstract

We study the relation between the total variation (TV) and Hellinger distances between two Gaussian location mixtures. Our first result establishes a general upper bound: for any two mixing distributions supported on a compact set, the Hellinger distance between the two mixtures is controlled by the TV distance raised to a power $1 - o(1)$, where the $o(1)$ term is of order $1/\log\log(1/\mathrm{TV})$. We also construct two sequences of mixing distributions that demonstrate the sharpness of this bound. Taken together, our results resolve an open problem raised in Jia et al. (2023) and thus lead to an entropic characterization of learning Gaussian mixtures in total variation. Our inequality also yields optimal robust estimation of Gaussian mixtures in Hellinger distance, which has a direct implication for bounding the minimax regret of empirical Bayes under Huber contamination.

## 1. Introduction

The Gaussian location mixture model is one of the most fundamental models in nonparametric density estimation, Bayesian inference, and clustering (Lindsay, 1995; Dasgupta, 1999). Given a probability measure $\pi$ supported on $\mathbb{R}^d$, the corresponding Gaussian mixture density is defined as

$$f_\pi(x) := \int_{\mathbb{R}^d} \phi_d(x - \theta)\, d\pi(\theta),$$

where $\phi_d(x) := (2\pi)^{-d/2} \exp(-\|x\|_2^2/2)$ denotes the density of the $d$-dimensional standard Gaussian distribution.

In this paper, we study the relation between the total variation distance $\mathrm{TV}(p, q) := \frac{1}{2}\int |p - q|$ and the Hellinger

[1]Department of Statistics, University of Chicago, Chicago, IL, United States. Correspondence to: Joonhyuk Jung <joonhyukjung@uchicago.edu>.

*Proceedings of the 43rd International Conference on Machine Learning*, Seoul, South Korea. PMLR 306, 2026. Copyright 2026 by the author(s).

distance $H(p, q) := \sqrt{\frac{1}{2}\int(\sqrt{p} - \sqrt{q})^2}$ between two Gaussian mixture densities. Without any restriction on the distributions, it is well known that

$$H^2(p, q) \leq \mathrm{TV}(p, q) \leq \sqrt{2}\, H(p, q). \tag{1}$$

The Hellinger distance is a commonly used loss function in density estimation (Wong & Shen, 1995). It is especially useful for Gaussian location mixture estimation due to its direct implication for bounding the regret of an empirical Bayes estimator based on a plug-in estimator of the prior (Jiang & Zhang, 2009). When the data contain a small fraction of arbitrary outliers, the density estimation problem can be viewed as misspecified under total variation. Therefore, sharp inequalities are necessary to derive optimal error rates for robust density estimation of Gaussian location mixtures, and the inequalities in (1) are too loose for this purpose.

Relations between $f$-divergences for Gaussian location mixtures have been studied in the literature. In particular, for distributions $\pi$ and $\eta$ supported on a bounded Euclidean ball $\{\theta \in \mathbb{R}^d : \|\theta\|_2 \leq M\}$, Jia et al. (2023) proved that the induced Gaussian mixtures $f_\pi$ and $f_\eta$ satisfy

$$H^2(f_\pi, f_\eta) \asymp \mathrm{KL}(f_\pi \| f_\eta), \tag{2}$$

up to constant factors depending on $M$ and $d$. Here, $\mathrm{KL}(p\|q) := \int p \log \frac{p}{q}$ denotes the Kullback-Leibler divergence. The relation in (2) implies an entropic characterization of the minimax rate for estimating Gaussian location mixtures. The paper Jia et al. (2023) also investigated the relation between the total variation distance $\mathrm{TV}(f_\pi, f_\eta)$ and the $L_2$ distance $\|f_\pi - f_\eta\|_2$. However, whether the relation $\mathrm{TV}(f_\pi, f_\eta) \asymp H(f_\pi, f_\eta)$ holds was explicitly posed as an open question.

In this paper, we resolve this open problem by proving that

$$H(f_\pi, f_\eta) \leq \mathrm{TV}^{1-o(1)}(f_\pi, f_\eta), \tag{3}$$

where the $o(1)$ term in the exponent is of order

$$\frac{\Theta(1)}{\log\log(1/\mathrm{TV}(f_\pi, f_\eta))}.$$

We also construct sequences of distributions $\pi_n$ and $\eta_n$ showing that the $o(1)$ term is indeed necessary, thereby disproving the relation $\mathrm{TV}(f_\pi, f_\eta) \asymp H(f_\pi, f_\eta)$ for Gaussian location mixtures. Our proof is based on an expansion of the ratio $(f_\pi - f_\eta)/\phi_d$ in terms of Hermite polynomials. The key ingredients of the analysis are the derivation of a multivariate Nikolskii-type inequality (Proposition A.6) and a restricted-range inequality (Proposition A.7).

As a direct application, we show that for density estimation of $f_\pi$ under the Huber contamination model $(1-\epsilon)P_{f_\pi} + \epsilon Q$, where $Q$ is arbitrary, the minimax rate under the Hellinger distance is given by

$$\epsilon^{1 - \frac{\Theta(1)}{\log\log(1/\epsilon)}},$$

provided that the sample size satisfies $n \geq \mathrm{poly}(1/\epsilon)$.

### 1.1. Paper Organization

The remainder of this paper is organized as follows. Our main results are presented in Section 2, followed by the sharpness construction in Section 3. Two applications of the main results—an entropic characterization of Gaussian location mixture estimation in total variation and robust density estimation—are discussed in Section 4. In Section 5, we briefly discuss several open directions. Due to page limits, most technical proofs are deferred to the appendices.

### 1.2. Notation

Let $\mathbb{N}_0$ be the set of nonnegative integers and $\mathbb{R}$ the set of real numbers. We use the boldface notation, e.g., $\mathbf{k}$ and $\mathbf{l}$, for multi-index. For $\mathbf{k} = (k_1, \ldots, k_d) \in \mathbb{N}_0^d$, we write $|\mathbf{k}| := k_1 + \cdots + k_d$. We denote by $\|\theta\|_2$ and $\|\theta\|_\infty$ the Euclidean norm and $\infty$-norm of $\theta \in \mathbb{R}^d$, respectively. For a real matrix $A \in \mathbb{R}^{m \times n}$, $\|A\|_\infty := \max\{\|Ax\|_\infty : \|x\|_\infty = 1\}$ is the operator norm induced by the $\infty$-norm of vectors. Recall that $\phi_d$ denotes the $d$-dimensional standard Gaussian density. We may use $\phi = \phi_1$ when we only discuss one-dimensional results. For $p \in \{1, 2\}$, a measurable set $\mathcal{A} \subseteq \mathbb{R}^d$, and a measurable function $g : \mathbb{R}^d \to \mathbb{R}$, we write $\|g\|_{L^p(\mathcal{A}, \phi_d)}$ as $\left(\int_\mathcal{A} |g(x)|^p \phi_d(x)\, dx\right)^{1/p} = \left(\int_\mathcal{A} |g|^p \phi_d\right)^{1/p}$, whenever the integral exists. The abbreviation for $L^p(\mathbb{R}^d, \phi_d)$ is often $L^p(\phi_d)$ when no confusion arises. Let $\Pi_n^d$ be the set of real polynomials of total degree $\leq n$ in $d$ variables. We also write $\Pi_n = \Pi_n^1$ when $d = 1$. For $k \in \mathbb{N}_0$, we define the one-dimensional (normalized) Hermite polynomial $h_k \in \Pi_k$ by

$$h_k(x) := \frac{(-1)^k}{\sqrt{k!}\,\phi(x)} \frac{d^k}{dx^k} \phi(x). \tag{4}$$

For arbitrary dimensions, we define the Hermite polynomial $h_\mathbf{k} \in \Pi_{|\mathbf{k}|}^d$ by tensor products of one-dimensional Hermite polynomials:

$$h_\mathbf{k}(x) := \prod_{j=1}^d h_{k_j}(x_j).$$

Note that $\deg h_\mathbf{k} = |\mathbf{k}|$ and the collection $\{h_\mathbf{k} : |\mathbf{k}| \leq n\}$ forms an orthonormal basis of $\Pi_n^d$ with respect to the $L^2(\phi_d)$-norm. The dimension of $\Pi_n^d$ is given by $\binom{n+d}{n}$. For integer or real values, we write $a \vee b := \max\{a, b\}$ and $a \wedge b := \min\{a, b\}$. For a positive integer $N \in \mathbb{N}$, we write $[N] := \{1, \ldots, N\}$. For a real number $x$, $\lceil x \rceil$ is the smallest integer no smaller than $x$ and $\lfloor x \rfloor$ is the largest integer no larger than $x$. For $a, b : \mathcal{G} \to [0, \infty)$, we write $a \lesssim b$ or $a = O(b)$ if there exists some constant $C > 0$ independent of $g$ such that $a(g) \leq Cb(g)$ holds for all $g \in \mathcal{G}$. We write $a \gtrsim b$ or $a = \Omega(b)$ if $b \lesssim a$. We write $a \asymp b$ or $a = \Theta(b)$ if $a \lesssim b$ and $b \lesssim a$.

## 2. Main Results

In this section, we present our main results. The first result bounds the $\chi^2$-divergence $\chi^2(p\|q) := \int \frac{(p-q)^2}{q}$ of Gaussian mixtures in terms of the total variation distance, which immediately implies (3) since $H^2(p, q) \leq \chi^2(p\|q)$ holds in general.

**Theorem 2.1** (Inequality between TV distance and $\chi^2$-divergence). *Let $\pi$ and $\eta$ be probability measures supported on the $d$-dimensional cube $[-M, M]^d$. Let $\delta > 0$. Then, there exists $C_0 = C_0(\delta, M, d) > 0$, not depending on $\pi$ or $\eta$, such that*

$$\sqrt{\chi^2(f_\pi\|f_\eta)}$$
$$\leq \left(C_0 \vee \mathrm{TV}^{-\alpha(\mathrm{TV}(f_\pi, f_\eta))}(f_\pi, f_\eta)\right) \mathrm{TV}(f_\pi, f_\eta),$$

*where we define*

$$\alpha(t) := \frac{2 + \delta}{\log\left(\log(1/t) \vee e\right)}, \tag{5}$$

*for $t > 0$.*

*Remark* 2.2. Note that $\alpha(t)$ is increasing in $t$ and that $\alpha(t) \to 0$ as $t \downarrow 0$. However, $t^{-\alpha(t)}$ is decreasing in $t$ and $t^{-\alpha(t)} \to +\infty$ as $t \downarrow 0$.

*Remark* 2.3. We note that the exponent $\alpha(t)$ does not depend on $M$ or $d$. The dependence on $M$ and $d$ appears solely in the constant $C_0$. We defer the detailed discussion to Appendix A.3. In summary, $\log C_0$ exhibits a polynomial dependence on $M^2 d$, and becomes nearly linear when $\delta$ is large.

**Corollary 2.4** (Inequality between TV and Hellinger distances). *Let $\pi$ and $\eta$ be probability measures supported on the $d$-dimensional cube $[-M, M]^d$. Let $\delta > 0$. Then, there*

exists $C_0 = C_0(\delta, M, d) > 0$, *not depending on $\pi$ or $\eta$, such that*

$$H(f_\pi, f_\eta)$$
$$\leq \left( C_0 \vee \mathrm{TV}^{-\alpha(\mathrm{TV}(f_\pi, f_\eta))}(f_\pi, f_\eta) \right) \mathrm{TV}(f_\pi, f_\eta),$$

*where we define $\alpha(\cdot)$ as in* (5).

*Proof.* This is a direct consequence of Theorem 2.1 because $H^2(p, q) \leq \chi^2(p\|q)$ holds in general. □

A key step in establishing Theorem 2.1 and Corollary 2.4 is to relate the $L^1(\phi_d)$ and $L^2(\phi_d)$ norms of the ratio

$$g := \frac{f_\pi - f_\eta}{\phi_d}.$$

Indeed, the $L^1(\phi_d)$-norm of $g$ is exactly twice the total variation distance, while both the squared Hellinger distance and the $\chi^2$-divergence are closely related to the squared $L^2(\phi_d)$-norm.

A natural analysis is through a basis expansion in $L^2(\phi_d)$. In particular, the Hermite polynomial expansion of $g$ plays a central role, since its coefficients are precisely the moment differences between $\pi$ and $\eta$ (see Lemma A.1). Moreover, in the one-dimensional setting ($d = 1$), inequalities relating the $L^1(\phi_d)$ and $L^2(\phi_d)$ norms on finite-dimensional subspaces—such as Nikolskii-type and restricted-range inequalities—have been extensively studied (Lubinsky, 2007).

**Theorem 2.5** (Inequality between $L^1(\phi_d)$ and $L^2(\phi_d)$ norms). *Let $\pi$ and $\eta$ be probability measures supported on the $d$-dimensional cube $[-2M, 2M]^d$. Define $g := \frac{f_\pi - f_\eta}{\phi_d}$ and suppose $\delta > 0$. Then, there exists $C_0 = C_0(\delta, M, d) > 0$, not depending on $\pi$ or $\eta$, such that*

$$\|g\|_{L^2(\phi_d)}$$
$$\leq \left( C_0 \vee \mathrm{TV}^{-\alpha(\mathrm{TV}(f_\pi, f_\eta))}(f_\pi, f_\eta) \right) \mathrm{TV}(f_\pi, f_\eta),$$

*where we define $\alpha(\cdot)$ as in* (5).

*Proof.* Here we give a sketch of the proof for the one-dimensional setting with $d = 1$. We provide the full proof for general $d$ in Appendix A.

Recall the definition (4) of the (one-dimensional) Hermite polynomials, and consider the following Hermite polynomial expansion (see Lemma A.1) of $g$.

$$g(x) = \int_{\mathbb{R}} \frac{\phi_1(x - \theta)}{\phi_1(x)} \, d(\pi - \eta)(\theta)$$
$$= \int_{\mathbb{R}} \sum_{k=0}^{\infty} \frac{\theta^k}{\sqrt{k!}} h_k(x) \, d(\pi - \eta)(\theta) \quad \text{(by Lemma A.1)}$$
$$= \sum_{k=0}^{\infty} \frac{\Delta_k}{\sqrt{k!}} h_k(x),$$

where $\Delta_k := \int_{\mathbb{R}} \theta^k \, d(\pi - \eta)(\theta)$. We decompose $g = q + r$, where

$$q = \sum_{k=0}^{n} \frac{\Delta_k}{\sqrt{k!}} h_k, \qquad r = \sum_{k=n+1}^{\infty} \frac{\Delta_k}{\sqrt{k!}} h_k,$$

and $n$ is an integer to be determined later. That is, $q$ is the $L^2(\phi_1)$ projection of $g$ onto a finite-dimensional subspace. To control the $L^1(\phi_1)$-norm of $q \in \Pi_n$, we define

$$c_n := \inf \left\{ \|P\|_{L^1(\phi_1)} : P \in \Pi_n, \|P\|_{L^2(\phi_1)} = 1 \right\}. \quad (6)$$

Note first that $c_n \leq 1$ holds by the Cauchy-Schwarz inequality. For $P \in \Pi_n$, the Nikolskii-type inequality (Nevai & Totik, 1987) states that

$$\sup_{x \in \mathbb{R}} \left| P(x) \phi_1^{1/2}(x) \right| \lesssim n^{1/4} \|P\|_{L^2(\phi_1)}. \quad (7)$$

We next show that $c_n \geq cn^{-1/4}e^{-n}$ holds for some universal constant $c > 0$:

$$\|P\|_{L^2(\phi_1)}^2$$
$$= \int_{-\infty}^{\infty} P^2 \phi_1$$
$$\leq 2 \int_{-2\sqrt{n+1}}^{2\sqrt{n+1}} P^2 \phi_1 \qquad \text{(Restricted-range inequality)}$$
$$\leq 2 \sup_{|x| \leq 2\sqrt{n+1}} \left| \phi_1^{-1/2}(x) \right| \sup_{x \in \mathbb{R}} \left| P(x) \phi_1^{1/2}(x) \right| \int_{-\infty}^{\infty} |P\phi_1|$$
$$\lesssim e^n \cdot n^{1/4} \|P\|_{L^2(\phi_1)} \cdot \|P\|_{L^1(\phi_1)}. \qquad \text{(by (7))}$$

The restricted-range inequality used above follows from Theorem 6.2(b) of Lubinsky (2007) with $W = \phi_1^{1/2}$ being a Freud-type weight function.

In addition to $c_n$, another technical ingredient is to control the tail norm $\|r\|_{L^2(\phi_1)}$. Compact support implies $|\Delta_k| \leq 2(2M)^k$ and

$$\|r\|_{L^2(\phi_1)} \leq \left( \sum_{k=n+1}^{\infty} \frac{4(4M^2)^k}{k!} \right)^{1/2} \leq \left( \frac{C}{n+1} \right)^{(n+1)/2},$$

where $C$ is a positive constant depending solely on $M$.

Now we bound from below $\|g\|_{L^1(\phi_1)}$ as follows.

$$\|g\|_{L^1(\phi_1)} \geq \|q\|_{L^1(\phi_1)} - \|r\|_{L^1(\phi_1)}$$
$$\geq c_n \|q\|_{L^2(\phi_1)} - \|r\|_{L^2(\phi_1)} \qquad \text{(by (6))}$$
$$\geq c_n \|g\|_{L^2(\phi_1)} - 2\|r\|_{L^2(\phi_1)},$$

where the last inequality uses $c_n \leq 1$ and the decomposition $g = q + r$. Together with the lower bound on $c_n$ and the upper bound on $\|r\|_{L^2(\phi_1)}$, we obtain

$$2t \geq \sup_{n \geq 1} \left\{ cn^{-1/4}e^{-n} \|g\|_{L^2(\phi_1)} - 2\left( \frac{C}{n+1} \right)^{(n+1)/2} \right\},$$

where $t = \frac{1}{2}\|g\|_{L^1(\phi_1)} = \mathrm{TV}(f_\pi, f_\eta)$. Finally, we choose

$$n \approx \frac{2\log(1/t)}{\log\log(1/t)}$$

to conclude the proof. Later, in Appendix A.1, we present multidimensional extensions of the Nikolskii-type and restricted-range inequalities (Propositions A.6 and A.7). Building on these results, we provide the full proof of the theorem in Appendix A.2. □

*Proof of Theorem 2.1.* Here we show that Theorem 2.1 follows from Theorem 2.5 and that the constants $C_0$ in both theorems coincide. Fix $\theta \in [-M, M]^d$. Consider the translation map $\tau_\theta(x) = x - \theta$ and define the following push-forward measures:

$$\pi_\theta := (\tau_\theta)_\sharp \pi, \qquad \eta_\theta := (\tau_\theta)_\sharp \eta.$$

Note that these are simply translations of the original measures and are supported on $[-2M, 2M]^d$. Define $g_\theta := \frac{f_{\pi_\theta} - f_{\eta_\theta}}{\phi_d}$. Then,

$$\|g_\theta\|_{L^2(\phi_d)}^2 = \int_{\mathbb{R}^d} \frac{(f_\pi(x+\theta) - f_\eta(x+\theta))^2}{\phi_d(x)}\,dx$$
$$= \int_{\mathbb{R}^d} \frac{(f_\pi(x) - f_\eta(x))^2}{\phi_d(x-\theta)}\,dx,$$
$$\|g_\theta\|_{L^1(\phi_d)} = \int_{\mathbb{R}^d} |f_\pi(x+\theta) - f_\eta(x+\theta)|\,dx$$
$$= \int_{\mathbb{R}^d} |f_\pi(x) - f_\eta(x)|\,dx = 2\,\mathrm{TV}(f_\pi, f_\eta).$$

Since $g_\theta$ obeys the inequality in Theorem 2.5, there exists $C_0 = C_0(\delta, M, d) > 0$, not depending on $\pi$, $\eta$, or $\theta$, such that

$$\left(\int \frac{(f_\pi(x) - f_\eta(x))^2}{\phi_d(x-\theta)}\,dx\right)^{1/2}$$
$$\leq \left(C_0 \vee \mathrm{TV}^{-\alpha(\mathrm{TV}(f_\pi, f_\eta))}(f_\pi, f_\eta)\right)\mathrm{TV}(f_\pi, f_\eta).$$

Meanwhile, we can apply Jensen's inequality pointwise in $x$ to get

$$\frac{(f_\pi(x) - f_\eta(x))^2}{f_\eta(x)} \leq \int \frac{(f_\pi(x) - f_\eta(x))^2}{\phi_d(x-\theta)}\,d\eta(\theta).$$

Integrate both sides in $x$. Then, use Fubini-Tonelli (nonnegativity) and the fact that a mixture integral is upper bounded by the supremum of its integrand to show that

$$\chi^2(f_\pi\|f_\eta) \leq \sup_{\theta \in [-M,M]^d} \int \frac{(f_\pi(x) - f_\eta(x))^2}{\phi_d(x-\theta)}\,dx,$$

thus concluding the proof. □

## 3. Sharpness

This section establishes the sharpness of the inequalities in the preceding section. Concretely, Theorem 3.1 demonstrates the sharpness of Corollary 2.4; consequently, the sharpness of Theorem 2.1 follows immediately from the inequality $H^2(p, q) \leq \chi^2(p\|q)$. We prove, by construction, that the exponent $\alpha(\cdot)$ is necessary up to a universal constant. Since the construction is one-dimensional, we write $\phi = \phi_1$ throughout this section for simplicity of notation.

**Theorem 3.1** (Sharpness of Corollary 2.4). *There exist two sequences of probability measures $\{\pi_n\}$ and $\{\eta_n\}$ supported on $[-M, M]$ such that, if we define*

$$\mathrm{TV}_n := \mathrm{TV}(f_{\pi_n}, f_{\eta_n}), \qquad H_n := H(f_{\pi_n}, f_{\eta_n}),$$

*then $\mathrm{TV}_n \downarrow 0$ as $n \to \infty$, and moreover it holds for all $n$ that $\mathrm{TV}_n < e^{-e}$ and that*

$$H_n \geq \mathrm{TV}_n^{1-\alpha^*(\mathrm{TV}_n)},$$

*where we define*

$$\alpha^*(t) := \frac{0.33}{\log\log(1/t)}, \qquad t > 0.$$

To prove Theorem 3.1, we first construct three pairs of sequences of mixing distributions: $(\pi_n^{(0)}, \eta_n^{(0)})$, $(\pi_n^{(1)}, \eta_n^{(1)})$, and $(\pi_n^{(2)}, \eta_n^{(2)})$.

$$\underset{\text{Lemma 3.2}}{\begin{bmatrix} \pi_n^{(0)} \\ \eta_n^{(0)} \end{bmatrix}} \overset{(15)}{\underset{}{\mapsto}} \underset{\text{Corollary 3.3}}{\begin{bmatrix} \pi_n^{(1)} \\ \eta_n^{(1)} \end{bmatrix}} \overset{(16)}{\underset{}{\mapsto}} \underset{\text{Corollary 3.4}}{\begin{bmatrix} \pi_n^{(2)} \\ \eta_n^{(2)} \end{bmatrix}} \overset{(18)}{\underset{}{\mapsto}} \underset{\text{Theorem 3.1}}{\begin{bmatrix} \pi_n \\ \eta_n \end{bmatrix}}$$

First, the pair $(\pi_n^{(0)}, \eta_n^{(0)})$, constructed in Lemma 3.2, provides a sharp example of the inequality between the $L^1(\phi)$ and $L^2(\phi)$ norms (Theorem 2.5). Corollary 3.3 then modifies this construction to obtain $(\pi_n^{(1)}, \eta_n^{(1)})$, for which a lower bound on the $\chi^2$-divergence is available. Next, Corollary 3.4 further transforms $(\pi_n^{(1)}, \eta_n^{(1)})$ into $(\pi_n^{(2)}, \eta_n^{(2)})$, yielding a lower bound on the Hellinger distance. Combining these results, we complete the proof of Theorem 3.1 at the end of this section.

Before constructing the sharp example $(\pi_n^{(0)}, \eta_n^{(0)})$ of Theorem 2.5, we recall the key ingredients of its proof: (1) the quantity $c_n$, defined in (6), admits a lower bound of the form $e^{-O(n)}$; and (2) the tail norm $\|r\|_{L^2(\phi)}$ can be controlled by $e^{-\Omega(n\log n)}$ via differences in higher-order moments of the mixing distributions.

We also note that the monomials $(x^n)_n$ provide a sharp instance for $c_n$, since the norm ratio $\|x^n\|_{L^1(\phi)} / \|x^n\|_{L^2(\phi)}$ decays exponentially in $n$. Motivated by this, for a given $n$, we construct an example such that the $L^2(\phi)$ projection of

$(f_{\pi_n} - f_{\eta_n})/\phi$ onto $\Pi_n$ is proportional to $x^n$. To this end, we first choose $(n + 1)$ points in $[-M, M]$ as the support of the mixing distributions, denoted by $\theta_0, \ldots, \theta_n$, and then match the lower-order moments $\Delta_0, \ldots, \Delta_n$ so that

$$\sum_{k=0}^{n} \frac{\Delta_k}{\sqrt{k!}} h_k \propto x^n.$$

Given the values of $\theta_0, \ldots, \theta_n$, the differences in the lower-order moments $\Delta_0, \ldots, \Delta_n$ can be determined by solving a linear system involving the inversion of a Vandermonde matrix (see Lemma B.2 for its definition). We choose $\theta_0, \ldots, \theta_n$ to be the zeros of the $(n + 1)$-th Chebyshev polynomial of the first kind (i.e., Chebyshev nodes), since they lie in $[-1, 1]$ and yield a well-conditioned Vandermonde matrix (Gautschi, 1974). The $(n + 1)$-th Chebyshev polynomial of the first kind, denoted by $T_{n+1} \in \Pi_{n+1}$, is defined by

$$T_{n+1}(\cos(\theta)) = \cos((n + 1)\theta). \quad (8)$$

In addition to the boundedness of the nodes and the stability of the associated inverse Vandermonde matrix, another advantage of using Chebyshev nodes is that they enable recursive control of higher-order moments in terms of lower-order ones. The properties of this construction are summarized in Lemma 3.2, whose full proof is given in Appendix B.

**Lemma 3.2** (Sharp example of Theorem 2.5). *Let $n \geq 11$ be an odd integer and $\theta_j = \cos\left(\frac{2j+1}{2n+2}\pi\right), j = 0, \ldots, n$ be the zeros of Chebyshev polynomial of the first kind, $T_{n+1}(x)$. Given $M > 0$, define $a = 1 \wedge M$ and*

$$\Delta_k = \begin{cases} \dfrac{\left\{a(\sqrt{2} - 1)\right\}^{n+1}}{(n - k)!!}, & k \text{ is odd}, \\ 0, & k \text{ is even}, \end{cases}$$

*for $k = 0, 1, \ldots, n$, where $(n - k)!!$ is the double factorial. Define $(w_0, \ldots, w_n) \in \mathbb{R}^{n+1}$ to be the unique vector solving*

$$\Delta_k = \sum_{j=0}^{n} w_j (a\theta_j)^k, \qquad k = 0, 1, \ldots, n.$$

*Accordingly, define two discrete probability measures*

$$\pi_n^{(0)} := \sum_{j=0}^{n} \left(\frac{1}{n+1} + w_j\right) \delta_{a\theta_j},$$

$$\eta_n^{(0)} := \sum_{j=0}^{n} \frac{1}{n+1} \delta_{a\theta_j}, \quad (9)$$

*where $\delta_{a\theta_j}$ denotes the point mass at $a\theta_j$. Then,*

1. *$w_j$ is well-defined and $|w_j| \leq \frac{1}{n+1}$ for all $j$.*

2. *$\pi_n^{(0)}$ and $\eta_n^{(0)}$ are valid discrete probability measures supported on $[-M, M]$.*

3. *For $0 \leq k \leq n$, $\Delta_k = \int \theta^k d(\pi_n^{(0)} - \eta_n^{(0)})(\theta)$ satisfies*

$$|\Delta_k| \leq \left\{a(\sqrt{2} - 1)\right\}^{n+1} \exp\left(\frac{n}{5.54}\right) b^{k-n}, \quad (10)$$

*where $b := a\sqrt{\frac{n}{2.77}}$.*

4. *If we further define $\Delta_k := \int \theta^k d(\pi_n^{(0)} - \eta_n^{(0)})(\theta)$ for $k > n$, then (10) is also true.*

5. *If we write $q_n(x) = \sum_{k=0}^{n} \frac{\Delta_k}{\sqrt{k!}} h_k(x)$ and $r_n(x) = \sum_{k=n+1}^{\infty} \frac{\Delta_k}{\sqrt{k!}} h_k(x)$, then*

$$q_n(x) = \left\{a(\sqrt{2} - 1)\right\}^{n+1} \frac{x^n}{n!}.$$

*In addition, there exists a universal $N_0 \in \mathbb{N}$ such that it holds for all $n \geq N_0$ that*

$$\|r_n\|_{L^2(\phi)}$$
$$\leq \frac{1}{32} \exp\left(\frac{n}{5.53}\right) \|q_n\|_{L^1(\phi)} \quad (11)$$
$$\leq \frac{1}{16} \exp\left(-\left\{\frac{\log(2)}{2} - \frac{1}{5.53}\right\} n\right) \|q_n\|_{L^2(\phi)}. \quad (12)$$

6. *$g_n = q_n + r_n$ satisfies*

$$\lim_{n \to \infty} \frac{1}{n \log n} \log\left(\frac{1}{\|g_n\|_{L^1(\phi)}}\right)$$
$$= \lim_{n \to \infty} \frac{1}{n \log n} \log\left(\frac{1}{\|g_n\|_{L^2(\phi)}}\right)$$
$$= \frac{1}{2}. \quad (13)$$

*Proof.* We will give the full proof in Appendix B.2. The key argument, which is to derive the bound (10) for $k > n$ is sketched below. Write the Chebyshev polynomial as $T_{n+1}(x) = 2^n(x^{n+1} - \sigma_2 x^{n-1} + \sigma_4 x^{n-3} - \cdots + (-1)^{(n+1)/2}\sigma_{n+1})$. The choice of the support $\{a\theta_0, \ldots, a\theta_n\}$ implies that $T_{n+1}(\theta_j) = 0$ for all $j = 0, \cdots, n$, and thus $(a\theta_j)^{K+1} = \sigma_2 a^2(a\theta_j)^{K-1} - \sigma_4 a^4(a\theta_j)^{K-3} + \cdots + (-1)^{(n-1)/2}\sigma_{n+1}a^{n+1}(a\theta_j)^{K-n}$. This implies $|\Delta_{K+1}| = \left|\sum_{j=0}^{n} w_j(a\theta_j)^{K+1}\right| \leq \sigma_2 a^2|\Delta_{K-1}| + \sigma_4 a^4|\Delta_{K-3}| + \cdots + \sigma_{n+1}a^{n+1}|\Delta_{K-n}|$, from which we can bound all $|\Delta_k|$ for $k > n$ via mathematical induction. $\square$

**Corollary 3.3** (Sharp example of Theorem 2.1). *Recall the definition* (9) *of* $\pi_n^{(0)}$ *and* $\eta_n^{(0)}$ *from the above. Let*

$$R_n = \sqrt{8n+4}, \qquad \lambda_n = \exp(-R_n), \qquad (14)$$

*and accordingly define*

$$\pi_n^{(1)} := (1-\lambda_n)\delta_0 + \lambda_n \pi_n^{(0)},$$
$$\eta_n^{(1)} := (1-\lambda_n)\delta_0 + \lambda_n \eta_n^{(0)}, \qquad (15)$$

*where $\delta_0$ denotes the point mass at zero. Then, there exists a universal $N_0 \in \mathbb{N}$ such that it holds for all $n \geq N_0$ that*

$$\mathrm{TV}\left(f_{\pi_n^{(1)}}, f_{\eta_n^{(1)}}\right) = \frac{\lambda_n}{2}\|g_n\|_{L^1(\phi)},$$
$$\sqrt{\chi^2\left(f_{\pi_n^{(1)}} \| f_{\eta_n^{(1)}}\right)} \geq \frac{\lambda_n}{4}\|q_n\|_{L^2(\phi)}.$$

*Proof.* See Appendix B.2. □

**Corollary 3.4** (Sharp example of Corollary 2.4). *Recall the definition* (15) *of* $\pi_n^{(1)}$ *and* $\eta_n^{(1)}$ *from the above. Let*

$$\pi_n^{(2)} := \frac{1}{4}\pi_n^{(1)} + \frac{3}{4}\eta_n^{(1)}, \qquad \eta_n^{(2)} := \eta_n^{(1)}. \qquad (16)$$

*Then, there exists a universal $N_0 \in \mathbb{N}$ such that it holds for all $n \geq N_0$ that*

$$\mathrm{TV}\left(f_{\pi_n^{(2)}}, f_{\eta_n^{(2)}}\right) = \frac{\lambda_n}{8}\|g_n\|_{L^1(\phi)},$$
$$H\left(f_{\pi_n^{(2)}}, f_{\eta_n^{(2)}}\right) \geq \frac{\lambda_n}{64}\|q_n\|_{L^2(\phi)}. \qquad (17)$$

*Proof.* The equality for the total variation distance is straightforward. Now, observe for all $x \in \mathbb{R}$ that

$$u(x) := \frac{f_{\pi_n^{(1)}}(x)}{f_{\eta_n^{(1)}}(x)} - 1$$

$$= \frac{(1-\lambda_n)\phi(x) + \sum_{j=0}^{n}\left(\frac{\lambda_n}{n+1} + \lambda_n w_j\right)\phi(x - a\theta_j)}{(1-\lambda_n)\phi(x) + \sum_{j=0}^{n}\frac{\lambda_n}{n+1}\phi(x - a\theta_j)} - 1$$

$$\leq \max_{0 \leq j \leq n} \frac{\frac{\lambda_n}{n+1} + \lambda_n w_j}{\frac{\lambda_n}{n+1}} - 1 \leq 1 \qquad (\because |w_j| \leq \frac{1}{n+1})$$

and hence that $\|u\|_\infty \leq 1$. Write

$$H^2\left(f_{\pi_n^{(2)}}, f_{\eta_n^{(2)}}\right) = H^2\left(\frac{1}{4}f_{\pi_n^{(1)}} + \frac{3}{4}f_{\eta_n^{(1)}}, f_{\eta_n^{(1)}}\right)$$

$$= \int F\left(1 + \frac{u}{4}\right)f_{\eta_n^{(1)}},$$

where $F(t) := \frac{1}{2}(\sqrt{t} - 1)^2$. A Taylor expansion of $F$ gives

$$F\left(1 + \frac{u}{4}\right) = \frac{u^2}{128} - \frac{u^3}{32(4+v)^{5/2}} \quad \text{(for some } |v| \leq |u|)$$

$$\geq \frac{u^2}{128} - \frac{u^2}{288\sqrt{3}} \qquad (\|u\|_\infty \leq 1)$$

$$\geq \frac{u^2}{256}.$$

Integrating against $f_{\eta_n^{(1)}}$ yields

$$H^2\left(f_{\pi_n^{(2)}}, f_{\eta_n^{(2)}}\right) \geq \frac{1}{256}\chi^2\left(f_{\pi_n^{(1)}} \| f_{\eta_n^{(1)}}\right),$$

concluding the proof. □

Now we are ready to prove Theorem 3.1 (Sharpness of Corollary 2.4) with the above $(\pi_n^{(2)}, \eta_n^{(2)})$.

*Proof of Theorem 3.1.* Let

$$\mathrm{TV}_n := \mathrm{TV}\left(f_{\pi_n^{(2)}}, f_{\eta_n^{(2)}}\right), \quad H_n := H\left(f_{\pi_n^{(2)}}, f_{\eta_n^{(2)}}\right).$$

Then, (13), (14), and (17) imply that

$$\lim_{n \to \infty} \frac{1}{n \log n}\log\left(\frac{1}{\mathrm{TV}_n}\right) = \frac{1}{2}.$$

Thus, it holds for large enough $n$ that

$$8\|g_n\|_{L^1(\phi)} \leq 8\|q_n\|_{L^1(\phi)} + 8\|r_n\|_{L^2(\phi)}$$

$$\leq \frac{1}{2}\exp\left(\frac{n}{5.53}\right)\|q_n\|_{L^1(\phi)} \qquad \text{(by (11))}$$

$$\leq \exp\left(-\left\{\log(2) - \frac{2}{5.53}\right\}\frac{n}{2}\right)\|q_n\|_{L^2(\phi)}$$
$$\qquad\qquad\qquad\qquad\qquad \text{(by (12))}$$

$$\leq \exp\left(-0.33\frac{\log(1/\mathrm{TV}_n)}{\log\log(1/\mathrm{TV}_n)}\right)\|q_n\|_{L^2(\phi)}.$$

Multiply both sides by $\frac{\lambda_n}{64}$ to conclude that

$$\mathrm{TV}_n = \frac{\lambda_n}{8}\|g_n\|_{L^1(\phi)} \qquad \text{(by (17))}$$

$$\leq \mathrm{TV}_n^{\alpha^*(\mathrm{TV}_n)}\frac{\lambda_n}{64}\|q_n\|_{L^2(\phi)}$$
$$\qquad\qquad\qquad \text{(by the definition of } \alpha^*(\cdot))$$

$$\leq \mathrm{TV}_n^{\alpha^*(\mathrm{TV}_n)}H_n. \qquad \text{(again by (17))}$$

According to Lemma 3.2 and Corollaries 3.3 and 3.4, the above argument is valid for all odd integers $n \geq N_0$, where $N_0 \in \mathbb{N}$ is universal. We define

$$\pi_n := \pi_{2(n+N_0)+1}^{(2)}, \qquad \eta_n := \eta_{2(n+N_0)+1}^{(2)} \qquad (18)$$

to conclude the proof of Theorem 3.1. That is, we are relabeling indices via the map $n \mapsto 2(n+N_0)+1$. □

*Remark* 3.5. A careful reader can verify that the constant 0.33 in $\alpha^*(\cdot)$ can be replaced by any positive real strictly less than $\log(2) - \frac{1}{4\log(2)} \approx 0.332$.

# 4. Applications

In this section, we provide a couple of consequences of our results. The notations "$\lesssim, \gtrsim, \asymp$" in this section will hide constants depending on $M$ or $d$.

Due to page constraints, all the proofs for this section are deferred to Appendix C.

## 4.1. Entropic Characterization of Learning in TV

The characterization of minimax rates of estimation via metric entropy has been extensively studied (LeCam, 1973; Birgé, 1983; 1986; Yatracos, 1985; Haussler & Opper, 1997; Yang & Barron, 1999). While minimax upper and lower bounds do not necessarily match in general, recent work by Jia et al. (2023) showed that estimating Gaussian mixture densities with bounded support under the Hellinger distance admits an exact entropic characterization of the minimax rate, due to the fact that $H^2(f_\pi, f_\eta) \asymp \mathrm{KL}(f_\pi \| f_\eta)$. Similarly, our Corollary 2.4, which relates the total variation and Hellinger distances, implies a corresponding characterization for the same problem under total variation, up to a $1 - o(1)$ exponent in the rate.

We first define the metric entropy of Gaussian location mixtures, and then state a result of Jia et al. (2023).

**Definition 4.1.** Let $\mathcal{P}_{M,d}$ be the collection of $d$-dimensional Gaussian mixtures where the mixing distributions are supported on the $d$-dimensional cube $[-M, M]^d$. For a distribution class $\mathcal{P} \subseteq \mathcal{P}_{M,d}$, its (global) Hellinger covering number is defined by

$$N_H(\mathcal{P}, \epsilon) := \min \{N : \exists P_1, \ldots, P_N,$$
$$\sup_{R \in \mathcal{P}} \inf_{1 \le i \le N} H(R, P_i) \le \epsilon \}.$$

The local Hellinger covering number of $\mathcal{P}$ is

$$N_{H,loc}(\mathcal{P}, \epsilon) := \sup_{P \in \mathcal{P}, \eta \ge \epsilon} N_H(B_H(P, \eta), \eta/2),$$

where $B_H(P, \eta) = \{R \in \mathcal{P} : H(P, R) \le \eta\}$. We define the global/local total variation covering number in the same manner.

**Proposition 4.2** (Corollary 11 of Jia et al. (2023)). *Suppose $\mathcal{P}$ is a compact subset (in Hellinger) of $\mathcal{P}_{M,d}$. Let $\widehat{P} = \widehat{P}(X_1, \ldots, X_n)$ denote an estimator based on $X_1, \ldots, X_n$ drawn i.i.d. from $P \in \mathcal{P}$. Then,*

$$\inf_{\widehat{P}} \sup_{P \in \mathcal{P}} \mathbb{E}_P \left[ H^2 \left( P, \widehat{P} \right) \right]$$
$$\asymp \inf_{\widehat{P} \in \mathcal{P}} \sup_{P \in \mathcal{P}} \mathbb{E}_P \left[ H^2 \left( P, \widehat{P} \right) \right] \asymp \epsilon_n^2,$$

*where*

$$\epsilon_n^2 \asymp \inf_{\epsilon > 0} \left( \epsilon^2 + \frac{1}{n} \log N_{H,loc}(\mathcal{P}, \epsilon) \right). \quad (19)$$

Unlike the Hellinger distance, there only exists an entropic characterization of the minimax upper bound in total variation (Yatracos, 1985). An entropic lower bound is not available in the literature to the best of our knowledge. By Corollary 2.4, the rate $\epsilon_n$ determined by the local Hellinger entropy (19) also characterizes the minimax rate of estimation under total variation as follows.

**Theorem 4.3** (Learning Gaussian mixtures in total variation). *Under the same conditions as in Proposition 4.2, for any $\delta > 0$, we have*

$$\epsilon_n^{2\left(1 + \frac{2+\delta}{\log(\log(1/\epsilon_n) \vee e)}\right)} \lesssim \inf_{\widehat{P}} \sup_{P \in \mathcal{P}} \mathbb{E}_P \left[ \mathrm{TV}^2 \left( P, \widehat{P} \right) \right]$$
$$\asymp \inf_{\widehat{P} \in \mathcal{P}} \sup_{P \in \mathcal{P}} \mathbb{E}_P \left[ \mathrm{TV}^2 \left( P, \widehat{P} \right) \right]$$
$$\lesssim \epsilon_n^2,$$

*where we define $\epsilon_n$ as in (19).*

## 4.2. Robust Density Estimation

In this section, we consider the problem of estimating a Gaussian mixture from contaminated data,

$$X_1, \ldots, X_n \overset{i.i.d.}{\sim} P := (1 - \epsilon) P_{f_\pi} + \epsilon Q, \quad (20)$$

where the distribution $P_{f_\pi} \in \mathcal{P}_{M,d}$ has density $f_\pi$ and $Q$ is an arbitrary contamination distribution. The data-generating process in (20) is known as Huber's contamination model (Huber, 1964). Robust density estimation under Huber contamination has been previously studied by Liu & Gao (2019); Humbert et al. (2022); Zhang & Ren (2023), and kernel density estimators have been shown to achieve optimal rates for estimating Hölder smooth densities.

Our main goal is to estimate the Gaussian mixture $f_\pi$ under the Hellinger distance, since the Hellinger error in density estimation directly implies a regret bound for empirical Bayes learning in the Gaussian sequence model (Jiang & Zhang, 2009; Saha & Guntuboyina, 2020).

To this end, we first introduce a robust estimator that enjoys statistical guarantees under the total variation distance. This follows from the classical construction of Yatracos (1985), since the Huber contamination model (20) can be viewed as a special case of model misspecification measured in total variation distance. Details of the Yatracos' estimator are deferred to Appendix C.1. Although this estimator is not computationally efficient (i.e., not polynomial-time computable), it serves as a useful statistical benchmark. Its guarantee is stated in the following proposition.

**Proposition 4.4** (Robust density estimation in TV). *Consider the data-generating process in (20). Then, the Yatracos' estimator $\widetilde{f}$ satisfies*

$$\sup_{\pi, Q} \mathbb{E} \left[ \mathrm{TV}^2 \left( f_\pi, \widehat{f} \right) \right] \lesssim \epsilon^2 + \frac{\log^{d+1}(n)}{n},$$

*where the expectation is under* (20) *and the supremum is taken over all $Q$ and $\pi$ such that* $\mathrm{supp}(\pi) \subseteq [-M, M]^d$.

Note that the Yatracos' estimator is a proper estimator in the sense that $\widehat{f}$ itself is also a Gaussian location mixture with mixing distribution supported on $[-M, M]^d$. Thus, our Corollary 2.4 directly implies a minimax upper bound in Hellinger distance as follows.

**Theorem 4.5** (Robust density estimation in Hellinger). *Consider the data-generating process in* (20). *Suppose $\delta > 0$. Then, the Yatracos' estimator $\widehat{f}$ satisfies*

$$\sup_{\pi, Q} \mathbb{E}\left[ H^2\left(f_\pi, \widehat{f}\right) \right] \lesssim \mathcal{E}^2(\epsilon, n), \qquad (21)$$

*where we define*

$$\mathcal{E}^2(\epsilon, n) := \epsilon^{2\left(1 - \frac{2+\delta}{\log(\log(1/\epsilon)\vee e)}\right)} + \frac{1}{n^{1-o_d(1)}}, \qquad (22)$$

*the expectation is under* (20), *the supremum is taken over all $Q$ and $\pi$ such that* $\mathrm{supp}(\pi) \subseteq [-M, M]^d$, *and $o_d(1)$ is a positive real-valued function of $n$ and $d$, which converges to zero as $n \to \infty$.*

We note that estimation of $f_\pi$ in the Hellinger distance has previously been studied by Saha & Guntuboyina (2020); Kim & Guntuboyina (2022); Soloff et al. (2025) in the special case of (20) with $\epsilon = 0$. Compared with these results, the second term $n^{-(1-o_d(1))}$ in (22) may still be improvable. However, improving this term would require techniques different from those used in Corollary 2.4, and we leave this question for future work. On the other hand, the first term $\epsilon^{2\left(1 - \frac{2+\delta}{\log(\log(1/\epsilon)\vee e)}\right)}$ in (22) is optimal. The following result is obtained by applying the two-point argument of Chen et al. (2018) to the sharpness example constructed in Theorem 3.1.

**Theorem 4.6** (Minimax lower bound on robust density estimation in Hellinger). *Consider the data-generating process in* (20). *Then, we have*

$$\inf_{\widehat{f}} \sup_{\pi, Q} \mathbb{E}\left[ H^2\left(f_\pi, \widehat{f}\right) \right] \gtrsim \epsilon^{2\left(1 - \frac{0.33}{\log(\log(1/\epsilon)\vee e)}\right)},$$

*where the expectation is under* (20) *and the supremum is taken over all $Q$ and $\pi$ such that* $\mathrm{supp}(\pi) \subseteq [-M, M]^d$.

The Hellinger bound in Theorem 4.5 can be applied to empirical Bayes learning of Gaussian means with outliers. To motivate this application, consider the following Gaussian location model with prior $\pi$,

$$X \mid \theta \sim N(\theta, I_d), \qquad \theta \sim \pi.$$

Given knowledge of the prior, the (oracle) Bayes estimator under squared error loss is given by the posterior mean,

$$\widehat{\theta}^\star(X) = X + \frac{\nabla f_\pi(X)}{f_\pi(X)}. \qquad (23)$$

This formula is known as Tweedie's formula (Efron, 2011). Without knowledge of $\pi$, an empirical Bayes estimator replaces $f_\pi$ in (23) by its estimate,

$$\widehat{\theta}(X) := X + \frac{\nabla \widehat{f}(X)}{\widehat{f}(X)}.$$

The regret (Saha & Guntuboyina, 2020; Soloff et al., 2025) of $\widehat{\theta}(X)$ is quantified by

$$\mathbb{E}_{X \sim f_\pi} \left\| \widehat{\theta}(X) - \widehat{\theta}^\star(X) \right\|^2,$$

which is equal to the Fisher divergence between $f_\pi$ and $\widehat{f}$.

In a typical empirical Bayes setting, one has i.i.d. observations generated from $f_\pi$. Here, we consider a more general data-generating process in (20) that allows the presence of arbitrary outliers. This requires the estimator $\widehat{f}$ to be robust, and thus the Yatracos' estimator satisfying the risk bound in Theorem 4.5 is adopted here.

We note that the clean data setting of the problem with $\epsilon = 0$ has been well studied in the literature (James et al., 1961; Efron & Morris, 1972; 1973; Johnstone, 2002; Ignatiadis & Sen, 2025), and the nonparametric maximum likelihood estimator (NPMLE) and sieve MLE are shown to achieve the parametric rate up to some logarithmic factor (Wong & Shen, 1995; Genovese & Wasserman, 2000; Ghosal & Van der Vaart, 2001; Jiang & Zhang, 2009; Saha & Guntuboyina, 2020; Soloff et al., 2025). However, when $\epsilon > 0$, it is unclear whether the NPMLE still works in the presence of arbitrary outliers. We conjecture that the error rate of the NPMLE has a highly sub-optimal dependence on $\epsilon$.

In terms of techniques for analyzing the regret bound, results in Jiang & Zhang (2009); Saha & Guntuboyina (2020); Soloff et al. (2025) and related works crucially rely on controlling the Fisher divergence via the Hellinger distance. See, for instance, Theorem E.1 of Saha & Guntuboyina (2020). These works employ a regularized version of $\widehat{\theta}(X)$ to avoid numerical instability when the denominator is close to zero:

$$\widehat{\theta}_\rho(X) := X + \frac{\nabla \widehat{f}(X)}{\widehat{f}(X) \vee \rho}. \qquad (24)$$

Following the same strategy, Theorem 4.7 is an immediate consequence of Theorem 4.5.

**Theorem 4.7** (Robust regret bound). *Consider the data-generating process in* (20). *Suppose $\widehat{\theta}^\star(\cdot)$ is as in* (23). *Then, there exists $\rho = \rho(\epsilon, n) > 0$ such that $\widehat{\theta}_\rho(\cdot)$ in* (24) *with $\widehat{f}$ being the Yatracos' estimator satisfies*

$$\sup_{\pi, Q} \mathbb{E}\left[ \mathbb{E}_{X \sim f_\pi} \left\| \widehat{\theta}_\rho(X) - \widehat{\theta}^\star(X) \right\|^2 \right] \lesssim \mathcal{E}^2(\epsilon, n), \quad (25)$$

*where the outer expectation is under* (20)*, the supremum is taken over all Q and π such that* $\mathrm{supp}(\pi) \subseteq [-M, M]^d$*, and the error function* $\mathcal{E}^2(\epsilon, n)$ *is defined as in* (22)*.*

*Remark* 4.8. Recall that the Yatracos' estimator is proper, meaning that $\widehat{f} = f_{\widehat{\pi}}$ for some estimated mixing distribution $\widehat{\pi}$. In other words, the estimator naturally induces an estimator of the prior as well. Consequently, the above construction may also be interpreted as a form of G-modeling estimation in the sense of Efron (2014).

*Remark* 4.9. The dependence of the tuning parameter $\rho(\epsilon, n)$ on $\epsilon$ can be alleviated by the standard Lepskii's method (Lepskii, 1991; 1992) to achieve adaptive estimation when $\epsilon$ is unknown.

See Appendix C.2 for detailed proofs of Theorem 4.3, Proposition 4.4, Theorems 4.5, 4.6, and 4.7.

## 5. Discussion

We establish a sharp relation between the total variation and Hellinger distances in this paper. Our results are derived for $d$-dimensional isotropic Gaussian mixture models with fixed covariance $I_d$. While we discuss implications for empirical Bayes methods, these procedures often involve a joint prior on both location and covariance. Extending our results to heteroscedastic Gaussian mixtures is an interesting direction for future work. In addition, establishing a sharp connection between the total variation distance and the Fisher divergence would further deepen the understanding of empirical Bayes procedures in robust settings.

The scope of this paper is restricted to compactly supported Gaussian mixtures. It would be an interesting direction for future work to study the exact relation between total variation and Hellinger distances for sub-Gaussian mixing distributions as well as more general tail behaviors.

Another open problem closely related to this work is the sharp relation between the total variation and $L^2$ distances. Resolving this question would have direct implications for nonparametric density estimation under the $L^2$ loss.

We also note that Theorems 4.5 and 4.6 establish the minimax-optimal rate of robust density estimation in squared Hellinger distance with respect to the contamination level $\epsilon$. However, the optimal dependence on the sample size $n$ remains open. It is worth emphasizing that this is already a long-standing open problem even in the classical setting with $d = 1$ and $\epsilon = 0$ (Polyanskiy & Wu, 2021).

## Acknowledgements

The research of CG is supported in part by NSF Grants ECCS-2216912 and DMS-2310769, and an Alfred Sloan fellowship. The authors thank Nikolaos Ignatiadis for fruitful discussions on the implications of the paper's results for empirical Bayes methods. The authors also thank the four anonymous reviewers of ICML 2026 for their insightful feedback.

## Impact Statement

This paper presents work whose goal is to advance the field of Machine Learning. There are many potential societal consequences of our work, none of which we feel must be specifically highlighted here.

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

# A. Proof of the Main Results

## A.1. Preliminaries: Hermite Polynomials and Inequalities

This section has two main goals. The first is to develop an understanding of the Hilbert space $L^2(\mathbb{R}^d, \phi_d)$ through the Christoffel-Darboux (C-D) kernel (Proposition A.2), which paves the way for the proofs of the Nikolskii-type inequality (Proposition A.6) and restricted-range inequality (Proposition A.7). The second is to prove Proposition A.8, which is a key ingredient in the proof of our main result, Theorem 2.5.

The results in this section have important implications in quantum mechanics. However, we postpone their physical interpretation for the moment. We first proceed to prove Proposition A.8 and Theorem 2.5 without relying on physical intuition, and then return to discuss the physical meaning at the end.

The study of orthogonal polynomials has a long and rich history, ranging from the classical work of Szeg (1939) to the survey of Lubinsky (2007), among many others. Results on multivariate polynomials are relatively limited and scattered across diverse literatures, including theoretical mathematics and quantum physics, making a unified overview challenging. For the sake of keeping the present paper self-contained, we summarize the essential results in this section. We adopt the notation introduced in Section 1.2 and fix $d \geq 1$ throughout.

**Lemma A.1** (Hermite polynomial expansion). *For $\theta = (\theta_1, \ldots, \theta_d) \in \mathbb{R}^d$ and $x = (x_1, \ldots, x_d) \in \mathbb{R}^d$, we have*

$$\frac{\phi_d(x - \theta)}{\phi_d(x)} = \sum_{\mathbf{k} \in \mathbb{N}_0^d} \frac{\theta^{\mathbf{k}}}{\sqrt{\mathbf{k}!}} h_{\mathbf{k}}(x),$$

*where we define*

$$\theta^{\mathbf{k}} := \prod_{j=1}^d \theta_j^{k_j}, \qquad\qquad \mathbf{k}! := \prod_{j=1}^d k_j!.$$

*Proof.* The one-dimensional version of this identity is classical and easy to show. See, for example, Equation (5.5.7) of Szeg (1939). We can extend it to arbitrary dimensions as follows.

$$\begin{aligned}
\frac{\phi_d(x - \theta)}{\phi_d(x)} &= \exp\left( \langle \theta, x \rangle_2 - \frac{1}{2} \|\theta\|_2^2 \right) \\
&= \prod_{j=1}^d \exp\left( \theta_j x_j - \frac{1}{2} \theta_j^2 \right) \\
&= \prod_{j=1}^d \sum_{k_j=0}^\infty \frac{\theta_j^{k_j}}{\sqrt{k_j!}} h_{k_j}(x_j).
\end{aligned}$$

Expand the product to complete the proof. $\qquad\square$

**Proposition A.2** (Christoffel-Darboux kernel). *For $n \in \mathbb{N}_0$, define the $n$-th Christoffel-Darboux kernel $K_n$ by*

$$K_n(x, y) := \sum_{|\mathbf{k}| \leq n} h_{\mathbf{k}}(x) h_{\mathbf{k}}(y). \tag{26}$$

*Then, for every $x \in \mathbb{R}^d$,*

1. *$K_n(x, \cdot) \in \Pi_n^d$.*

2. *$\langle f, K_n(x, \cdot) \rangle_{L^2(\mathbb{R}^d, \phi_d)} = f(x)$ holds for all $f \in \Pi_n^d$.*

*Proof.* The first statement is obvious. Due to linearity, it suffices to prove the second statement when $f = h_{\mathbf{k}}$ for some $|\mathbf{k}| \leq n$, which follows immediately from the orthonormality of the Hermite basis. $\qquad\square$

**Proposition A.3** (Christoffel-Darboux function). *For every $x \in \mathbb{R}^d$,*

$$\inf \left\{ \|P\|^2_{L^2(\mathbb{R}^d, \phi_d)} : P \in \Pi_n^d, P(x) = 1 \right\} = \frac{1}{K_n(x, x)}. \tag{27}$$

*Proof.* For $P \in \Pi_n^d$ such that $P(x) = 1$, write $P = \sum_{|\mathbf{k}| \leq n} c_{\mathbf{k}} h_{\mathbf{k}}$ so that

$$1 = \left( \sum_{|\mathbf{k}| \leq n} c_{\mathbf{k}} h_{\mathbf{k}}(x) \right)^2 \leq \left( \sum_{|\mathbf{k}| \leq n} c_{\mathbf{k}}^2 \right) \left( \sum_{|\mathbf{k}| \leq n} h_{\mathbf{k}}^2(x) \right) = \|P\|^2_{L^2(\mathbb{R}^d, \phi_d)} K_n(x, x),$$

which proves the lower bound. The lower bound is attained by the polynomial $\frac{K_n(x, \cdot)}{K_n(x,x)} \in \Pi_n^d$ due to the reproducing property of the C-D kernel (Proposition A.2). $\qquad \square$

In view of Proposition A.3, it is important to study an upper bound on the diagonal entries $K_n(x, x)$ of the C–D kernel. To this end, we introduce a useful lemma.

**Lemma A.4** (Mehler's formula). *For $\mathbf{k} \in \mathbb{N}_0^d$, define $E_{\mathbf{k}} := 2|\mathbf{k}| + d$. For $x, y \in \mathbb{R}^d$ and $t > 0$, define the Mehler kernel by*

$$M(x, y; t) := \sum_{\mathbf{k} \in \mathbb{N}_0^d} e^{-t E_{\mathbf{k}}} h_{\mathbf{k}}(x) h_{\mathbf{k}}(y) \phi_d^{1/2}(x) \phi_d^{1/2}(y). \tag{28}$$

*Then, we have the following closed-form formula:*

$$M(x, y; t) = (4\pi \sinh(2t))^{-d/2} \exp \left( -\frac{\|x\|_2^2 + \|y\|_2^2}{4 \tanh(2t)} + \frac{\langle x, y \rangle_2}{2 \sinh(2t)} \right). \tag{29}$$

*If $y = x$, in particular, then*

$$M(x, x; t) = (4\pi \sinh(2t))^{-d/2} \exp \left( -\frac{\|x\|_2^2}{2} \tanh(t) \right). \tag{30}$$

*Proof.* The right hand side of (28) factorizes as

$$\prod_{j=1}^d \sum_{k_j \in \mathbb{N}_0} e^{-t(2k_j + 1)} h_{k_j}(x_j) h_{k_j}(y_j) \phi_1^{1/2}(x_j) \phi_1^{1/2}(y_j).$$

Since the closed-form formula (29) admits the same factorization, it suffices to show (29) only for $d = 1$. There are many known proofs of the one-dimensional Mehler's formula. One such proof dates back (at least) to Watson (1933). Since it is quite short, we include it below. Recall the Fourier transform of $\phi_1$:

$$\phi_1(x) = \frac{1}{2\pi} \int \exp \left( -\frac{\xi^2}{2} + ix\xi \right) d\xi.$$

Hence, from the definition of $h_k$,

$$\begin{aligned} h_k(x) \phi_1^{1/2}(x) &= \frac{(-1)^k}{\sqrt{k!}} \phi_1^{-1/2}(x) \frac{d^k}{dx^k} \phi_1(x) \\ &= \frac{1}{2\pi\sqrt{k!}} \phi_1^{-1/2}(x) \int (-i\xi)^k \exp \left( -\frac{\xi^2}{2} + ix\xi \right) d\xi. \end{aligned}$$

In conclusion,

$$\sum_{k=0}^{\infty} e^{-t(2k+1)} h_k(x) h_k(y) \phi_1^{1/2}(x) \phi_1^{1/2}(y)$$

$$= (2\pi)^{-3/2} \exp\left(-t + \frac{x^2 + y^2}{4}\right) \iint \exp\left(-\frac{\xi^2 + \zeta^2}{2} + ix\xi + iy\zeta\right) \sum_{k=0}^{\infty} \frac{(-e^{-2t}\xi\zeta)^k}{k!} \, d\xi d\zeta$$

$$= (2\pi)^{-3/2} \exp\left(-t + \frac{x^2 + y^2}{4}\right) \iint \exp\left(-\frac{\xi^2 + \zeta^2}{2} - e^{-2t}\xi\zeta + ix\xi + iy\zeta\right) \, d\xi d\zeta$$

$$= (2\pi(1 - e^{-4t}))^{-1/2} \exp\left(-t + \frac{x^2 + y^2}{4} - \frac{x^2 + y^2 - 2e^{-2t}xy}{2(1 - e^{-4t})}\right)$$

$$= (4\pi \sinh(2t))^{-1/2} \exp\left(-\frac{x^2 + y^2}{4\tanh(2t)} + \frac{xy}{2\sinh(2t)}\right).$$

Absolute convergence justifies exchanging the summation and integration. We have derived the explicit form of the Mehler's formula, which implies the following corollary. $\qquad\square$

**Corollary A.5** (Upper bounds of the C-D kernel). *Recall the definition* (26) *of Christoffel-Darboux kernel* $K_n(x, x)$. *For* $n \in \mathbb{N}_0$, *define*

$$E_{n,d} := 2n + d, \qquad\qquad C_{n,d} := \left(\frac{(n + d)^{n+d}}{n^n d^d}\right)^{1/2}. \qquad (31)$$

*Then, we have*

$$\sup_{x \in \mathbb{R}^d} K_n(x, x) \phi_d(x) \le (2\pi)^{-d/2} C_{n,d}, \qquad (32)$$

$$C_{n,d} \le \left(\frac{e(n + d)}{d}\right)^{d/2} = O(n^{d/2}). \qquad (33)$$

*Furthermore, for* $\kappa > 1$,

$$\int_{\|x\|_2 > \sqrt{2\kappa E_{n,d}}} K_n(x, x) \phi_d(x) \le \left(\frac{e}{2d}\sqrt{\frac{\kappa}{\kappa - 1}}\right)^{d/2} E_{n,d}^{d/2} \exp\left(-c(\kappa) E_{n,d}\right), \qquad (34)$$

*where we define* $c(\kappa) := \sqrt{\kappa(\kappa - 1)} - \log\left(\sqrt{\kappa} + \sqrt{\kappa - 1}\right) > 0$.

*Proof.* The inequality (33) is straightforward. The other inequalities (32) and (34) can be derived from the Chernoff bound using the Mehler's formula (Lemma A.4) as follows. For all $x \in \mathbb{R}^d$ and $t > 0$,

$$K_n(x, x) \phi_d(x) = \sum_{|\mathbf{k}| \le n} h_{\mathbf{k}}^2(x) \phi_d(x) \qquad\qquad \text{(by (26))}$$

$$\le e^{tE_{n,d}} \sum_{|\mathbf{k}| \le n} e^{-tE_{\mathbf{k}}} h_{\mathbf{k}}^2(x) \phi_d(x) \qquad\qquad (E_{\mathbf{k}} \le E_{n,d})$$

$$\le e^{tE_{n,d}} M(x, x; t) \qquad\qquad \text{(by (28))}$$

$$= e^{tE_{n,d}} (4\pi \sinh(2t))^{-d/2} \exp\left(-\frac{\|x\|_2^2}{2}\tanh(t)\right). \qquad\qquad \text{(by (30))}$$

Therefore,

$$\sup_{x \in \mathbb{R}^d} K_n(x, x) \phi_d(x) \le \inf_{t > 0} e^{tE_{n,d}} (4\pi \sinh(2t))^{-d/2} = (2\pi)^{-d/2} C_{n,d},$$

where the infimum is attained at $t = \frac{1}{4} \log \left(1 + \frac{d}{n}\right)$. Similarly, for all $t > 0$ and $0 < s < \tanh(t)$,

$$
\int_{\|x\|_2 > \sqrt{2\kappa E_{n,d}}} K_n(x, x) \phi_d(x)
$$

$$
\leq e^{t E_{n,d}} (4\pi \sinh(2t))^{-d/2} \int_{\|x\|_2 > \sqrt{2\kappa E_{n,d}}} \exp\left(-\frac{\|x\|_2^2}{2} \tanh(t)\right)
$$

$$
\leq \exp\left((t - \kappa s) E_{n,d}\right) (4\pi \sinh(2t))^{-d/2} \int_{\mathbb{R}^d} \exp\left(-\frac{\|x\|_2^2}{2}(\tanh(t) - s)\right)
$$

$$
= \exp\left((t - \kappa s) E_{n,d}\right) (2 \sinh(2t)(\tanh(t) - s))^{-d/2}.
$$

Now fix $t = \log\left(\sqrt{\kappa} + \sqrt{\kappa - 1}\right) > 0$ so that $\cosh(t) = \sqrt{\kappa}$ and that $\sinh(t) = \sqrt{\kappa - 1}$. Choose $s = \tanh(t) - \frac{d}{2\kappa E_{n,d}}$ so that

$$
\int_{\|x\|_2 > \sqrt{2\kappa E_{n,d}}} K_n(x, x) \phi_d(x) \leq \exp\left(\frac{d}{2} - c(\kappa) E_{n,d}\right) \left(\frac{2d\sqrt{\kappa(\kappa - 1)}}{\kappa E_{n,d}}\right)^{-d/2},
$$

which is the desired result. Note that the choice of $(t, s)$ is asymptotically optimal as $E_{n,d} \to \infty$. □

We have derived upper bounds on the diagonal entries $K_n(x, x)$ of the C-D kernel. Using these bounds, we now present three norm inequalities in $\Pi_n^d$, stated as Propositions A.6, A.7, and A.8.

The first is the Nikolskii-type inequality. In case $d = 1$, the Nikolskii-type inequality has been extensively studied. For instance, the paper by Nevai & Totik (1987) focuses on the one-dimensional setting and establishes the sharpness of the Nikolskii-type inequalities (with more general weight functions). Note that the Mhaskar–Rakhmanov–Saff (MRS) number $a_n$ discussed in that paper is linearly comparable to $\sqrt{2E_{n,d}}$, the threshold.

The second is the restricted-range inequality. Similarly, in the one-dimensional setting, the restricted-range inequality has been studied in great depth; see Chapter 6 of the survey Lubinsky (2007). In higher dimensions, a few results are known as well; for example, see Lemma 5 of Maizlish & Prymak (2015).

The third, to the best of our knowledge, does not have a standard name. It can, however, be derived as a combination of the preceding two and will play an essential role in our main result.

**Proposition A.6** (Nikolskii-type inequality). *Recall the definition* (31) *of* $C_{n,d}$. *For all* $P \in \Pi_n^d$, *we have*

$$
\sup_{x \in \mathbb{R}^d} \left| P(x) \phi_d^{1/2}(x) \right| \leq (2\pi)^{-d/4} C_{n,d}^{1/2} \|P\|_{L^2(\mathbb{R}^d, \phi_d)}.
$$

*Proof.* According to Proposition A.3 and Corollary A.5, it holds for all $x \in \mathbb{R}^d$ that

$$
P^2(x) \phi_d(x) \leq (2\pi)^{-d/2} C_{n,d} \frac{P^2(x)}{K_n(x, x)} \tag{by (32)}
$$

$$
\leq (2\pi)^{-d/2} C_{n,d} \|P\|_{L^2(\mathbb{R}^d, \phi_d)}^2. \tag{by (27)}
$$

Take square roots of the both sides to conclude the proof. □

**Proposition A.7** (Restricted-range inequality). *Recall the definition* (31) *of* $E_{n,d}$. *Suppose* $\kappa > 1$. *Then, there exists a constant* $A = A(\kappa)$, *depending only on* $\kappa$, *such that, if* $E_{n,d} \geq Ad$, *then, for all* $P \in \Pi_n^d$, *we have*

$$
\int_{\mathbb{R}^d} P^2 \phi_d \leq 2 \int_{\|x\|_2 \leq \sqrt{2\kappa E_{n,d}}} P^2(x) \phi_d(x).
$$

*Proof.* Suppose

$$
\frac{E_{n,d}}{d} \geq \frac{1}{c(\kappa)} \log\left(\frac{e}{c(\kappa)} \sqrt{\frac{\kappa}{\kappa - 1}} \vee e\right) =: A(\kappa), \tag{35}
$$

where we define $c(\kappa)$ as in Corollary A.5. For $P \in \Pi_n^d$, write $P = \sum_{|\mathbf{k}| \leq n} c_{\mathbf{k}} h_{\mathbf{k}}$ so that $\int P^2 \phi_d = \sum_{|\mathbf{k}| \leq n} c_{\mathbf{k}}^2$. We have

$$\int_{\|x\|_2 > \sqrt{2\kappa E_{n,d}}} P^2(x)\phi_d(x) = \sum_{|\mathbf{k}| \leq n} \sum_{|\mathbf{l}| \leq n} c_{\mathbf{k}} M_{\mathbf{k}\mathbf{l}} c_{\mathbf{l}},$$

where we define

$$M_{\mathbf{k}\mathbf{l}} := \int_{\|x\|_2 > \sqrt{2\kappa E_{n,d}}} h_{\mathbf{k}}(x) h_{\mathbf{l}}(x) \phi_d(x).$$

Here, $M = (M_{\mathbf{k}\mathbf{l}})$ is a $(\dim \Pi_n^d) \times (\dim \Pi_n^d)$ positive semi-definite matrix. Thus, its operator norm is bounded by its trace. That is,

$$\int_{\|x\|_2 > \sqrt{2\kappa E_{n,d}}} P^2(x)\phi_d(x) \leq \left( \int_{\mathbb{R}^d} P^2 \phi_d \right) \text{trace}(M).$$

It suffices to show that the trace is at most $\frac{1}{2}$. By the definition (26) of Christoffel-Darboux kernel,

$$\text{trace}(M) = \sum_{|\mathbf{k}| \leq n} M_{\mathbf{k}\mathbf{k}} = \int_{\|x\|_2 > \sqrt{2\kappa E_{n,d}}} K_n(x,x)\phi_d(x). \tag{36}$$

Note that $z \geq 2\log(a \vee e)$ implies $az \leq e^z$. Thus, the assumption (35) implies

$$\frac{e}{c(\kappa)} \sqrt{\frac{\kappa}{\kappa - 1}} \frac{2c(\kappa)}{d} E_{n,d} \leq \exp\left( \frac{2c(\kappa)}{d} E_{n,d} \right). \tag{37}$$

In conclusion,

$$\text{trace}(M) \leq \left( \frac{e}{2d} \sqrt{\frac{\kappa}{\kappa - 1}} E_{n,d} \right)^{d/2} \exp\left( -c(\kappa) E_{n,d} \right) \qquad \text{(by (34) and (36))}$$

$$\leq 2^{-d} \leq \frac{1}{2}. \qquad \text{(by (37))}$$

$\square$

The following Proposition A.8 is simply a combination of Propositions A.6 and A.7, and it plays a central role in the proof of our main result.

**Proposition A.8** (Asymptotic lower bound of $L^1(\mathbb{R}^d, \phi_d)$-norm in $\Pi_n^d$). *Recall the definition (31) of $E_{n,d}$ and $C_{n,d}$. Define*

$$c_{n,d} := \inf \left\{ \|P\|_{L^1(\mathbb{R}^d, \phi_d)} : P \in \Pi_n^d, \|P\|_{L^2(\mathbb{R}^d, \phi_d)} = 1 \right\}. \tag{38}$$

*If the assumption (35) of Proposition A.7 holds, then*

$$c_{n,d} \geq \frac{1}{2} C_{n,d}^{-1/2} e^{-\kappa E_{n,d}/2}. \tag{39}$$

*Proof.* For $P \in \Pi_n^d$,

$$\|P\|_{L^2(\mathbb{R}^d, \phi_d)}^2 \leq 2 \int_{\|x\|_2 \leq \sqrt{2\kappa E_{n,d}}} P^2(x)\phi_d(x) \qquad \text{(by Proposition A.7)}$$

$$\leq 2 \sup_{\|x\|_2 \leq \sqrt{2\kappa E_{n,d}}} \left| \phi_d^{-1/2}(x) \right| \sup_{x \in \mathbb{R}^d} \left| P(x)\phi_d^{1/2}(x) \right| \int_{\mathbb{R}^d} |P\phi_d|$$

$$\leq 2 \left( (2\pi)^{d/4} e^{\kappa E_{n,d}/2} \right) \left( (2\pi)^{-d/4} C_{n,d}^{1/2} \|P\|_{L^2(\mathbb{R}^d, \phi_d)} \right) \|P\|_{L^1(\mathbb{R}^d, \phi_d)}. \qquad \text{(by Proposition A.6)}$$

Cancel out $\|P\|_{L^2(\mathbb{R}^d, \phi_d)}$ from the both sides to prove the inequality (39). $\square$

**Corollary A.9.** *Recall the definition* (38) *of* $c_{n,d}$. *Suppose* $\kappa_1 > 1$. *Then, there exists a constant* $A_1 = A_1(\kappa_1)$, *depending only on* $\kappa_1$, *such that, if* $n \geq A_1 d$, *then we have* $c_{n,d} \geq 3e^{-\kappa_1 n}$.

*Proof.* Suppose

$$\frac{n}{d} \geq \inf_\kappa \left\{ 1 \vee \frac{A(\kappa)}{2} \vee \frac{1}{2(\kappa_1 - \kappa)} \log\left( \frac{3^8 e^{1+2\kappa}}{2(\kappa_1 - \kappa)} \vee e \right) \right\} =: A_1(\kappa_1), \tag{40}$$

where we define $A(\kappa)$ as in (35), and the infimum is taken with respect to $\kappa$ such that $1 < \kappa < \kappa_1$. Recall that $z \geq 2\log(a \vee e)$ implies $az \leq e^z$. Thus, the assumption (40) implies

$$\frac{3^8 e^{1+2\kappa}}{2(\kappa_1 - \kappa)} \frac{4(\kappa_1 - \kappa)}{d} n \leq \exp\left( \frac{4(\kappa_1 - \kappa)}{d} n \right). \tag{41}$$

In conclusion,

$$c_{n,d} \geq \frac{1}{2} C_{n,d}^{-1/2} e^{-\kappa E_{n,d}/2} \tag{by (39)}$$

$$\geq \frac{1}{2} \left( \frac{e^{1+2\kappa}(n+d)}{d} \right)^{-d/4} \exp\left(-\kappa n\right) \tag{by (33)}$$

$$\geq \frac{1}{3} \left( \frac{2e^{1+2\kappa}}{d} n \right)^{-d/4} \exp\left(-\kappa n\right) \tag{$\because n \geq d$}$$

$$\geq 3^{2d-1} \exp(-\kappa_1 n) \geq 3e^{-\kappa_1 n}. \tag{by (41)}$$

Since $E_{n,d} = 2n + d \geq 2n$, the assumption (40) also implies the assumption (35) of Proposition A.7. $\qquad\square$

We have derived all the preliminary results required for the proof of our main theorem. Lastly, we introduce one technical lemma to conclude this section.

**Lemma A.10** (Lambert W function)**.** *Given* $\kappa_2 > 1$, $B_0 \geq 1$, *and* $t \in (0,1)$, *define*

$$w_0 := 1 \vee \frac{2}{\kappa_2 - 1} \log\left( \frac{B_0}{\kappa_2 - 1} \vee e \right), \tag{42}$$

$$n_0 := \left\lfloor 2B_0 e^{w_0} \vee \frac{2\kappa_2 \log(1/t)}{\log\left(\log(1/t) \vee e\right)} \right\rfloor.$$

*Then, it holds for all* $n \geq n_0$ *that*

$$\left( \frac{2B_0}{n+1} \right)^{(n+1)/2} \leq t. \tag{43}$$

*Proof.* Let $w > 0$ be the unique positive real number such that $\log(1/t) = B_0 w e^w$. Then,

$$\left( \frac{2B_0}{2B_0 e^w} \right)^{B_0 e^w} = t.$$

Since the function $z \mapsto (2B_0/z)^{z/2}$ is decreasing for $z > 2B_0/e$, it suffices to show $n + 1 \geq 2B_0 e^w$ to prove the inequality (43). We divide the argument into two cases, (a) $w < w_0$ and (b) $w \geq w_0$. In case (a) $w < w_0$, it is obvious that $n + 1 \geq n_0 + 1 \geq 2B_0 e^{w_0} \geq 2B_0 e^w$. Hence, we now suppose (b) $w \geq w_0$. Recall that $z \geq 2\log(a \vee e)$ implies $az \leq e^z$. Thus, (42) implies

$$\frac{B_0}{\kappa_2 - 1}(\kappa_2 - 1)w \leq \exp\left((\kappa_2 - 1)w\right). \tag{44}$$

Furthermore, since $B_0 \geq 1$ and $w_0 \geq 1$, we have $\log(1/t) = B_0 w e^w \geq e$ and

$$n + 1 \geq \frac{2\kappa_2 \log(1/t)}{\log\left(\log(1/t) \vee e\right)} = \frac{2\kappa_2 B_0 w e^w}{\log\left(B_0 w e^w\right)} \geq 2B_0 e^w,$$

where the last inequality follows from (44). $\qquad\square$

## A.2. Proof of the Main Theorem

We have already shown in the main text that Theorem 2.5 implies Theorem 2.1. Therefore, we proceed to prove Theorem 2.5 here.

*Proof of Theorem 2.5.* Let $\kappa_1 > 1$ and $\kappa_2 > 1$ satisfy $2\kappa_1\kappa_2 = 2 + \delta$. First, in view of Corollary A.9, there exists a positive integer $A_1 = A_1(\kappa_1)$, depending only on $\kappa_1$, such that

$$n \geq A_1 d \implies c_{n,d} \geq 3e^{-\kappa_1 n}. \tag{45}$$

Let $t := \frac{1}{2} \|g\|_{L^1(\phi_d)} = \mathrm{TV}(f_\pi, f_\eta) \in (0, 1)$. In view of Lemma A.10, define

$$n := A_1 d \vee B \vee \left\lfloor \frac{2\kappa_2 \log(1/t)}{\log(\log(1/t) \vee e)} \right\rfloor \in \mathbb{N}_0, \tag{46}$$

where

$$B_0 = B_0(\kappa_1, M^2 d) := \left(1 \vee 2eM^2 d\right) e^{2\kappa_1}, \tag{47}$$

$$B = B(\kappa_1, \kappa_2, M^2 d) := \left\lfloor 2B_0 \exp\left(1 \vee \frac{2}{\kappa_2 - 1} \log\left(\frac{B_0}{\kappa_2 - 1} \vee e\right)\right) \right\rfloor. \tag{48}$$

Observe from Lemma A.1 that

$$g = \sum_{\mathbf{k} \in \mathbb{N}_0^d} \frac{\Delta_\mathbf{k}}{\sqrt{\mathbf{k}!}} h_\mathbf{k}, \qquad\qquad \Delta_\mathbf{k} = \int_{\mathbb{R}^d} \theta^\mathbf{k} d(\pi - \eta)(\theta).$$

We decompose $g = q + r$, where

$$q = \sum_{|\mathbf{k}| \leq n} \frac{\Delta_\mathbf{k}}{\sqrt{\mathbf{k}!}} h_\mathbf{k} \in \Pi_n^d, \qquad\qquad r = \sum_{|\mathbf{k}| > n} \frac{\Delta_\mathbf{k}}{\sqrt{\mathbf{k}!}} h_\mathbf{k}.$$

From the compactness of the support, $|\Delta_\mathbf{k}| \leq 2(2M)^{|\mathbf{k}|}$. Thus, by the multinomial theorem and Stirling's formula,

$$\sum_{|\mathbf{k}|=m} \frac{\Delta_\mathbf{k}^2}{\mathbf{k}!} \leq \sum_{|\mathbf{k}|=m} \frac{4(4M^2)^m}{\mathbf{k}!} = \frac{4(4M^2 d)^m}{m!} \leq \frac{4}{\sqrt{2\pi m}} \left(\frac{4eM^2 d}{m}\right)^m. \tag{49}$$

It follows from the definition (46) that $n + 1 \geq 2B_0 e \geq 2\left(1 \vee 2eM^2 d\right) e^{1+2\kappa_1} \geq 16 \vee 8eM^2 d$. Thus,

$$\begin{aligned} \|r\|_{L^2(\phi_d)}^2 = \sum_{|\mathbf{k}|>n} \frac{\Delta_\mathbf{k}^2}{\mathbf{k}!} &\leq \sum_{m=n+1}^\infty \frac{4}{\sqrt{2\pi(n+1)}} \left(\frac{4eM^2 d}{n+1}\right)^m & \text{(by (49))} \\ &\leq \sum_{m=n+1}^\infty \frac{1}{2^{m-n-1}\sqrt{2\pi}} \left(\frac{4eM^2 d}{n+1}\right)^{n+1} & (\because n+1 \geq 16 \vee 8eM^2 d) \\ &\leq \left(\frac{4eM^2 d}{n+1}\right)^{n+1}. & (\because 2 \leq \sqrt{2\pi}) \end{aligned}$$

It follows from the definition (47) of $B_0$ that $4eM^2 d \leq 2B_0 e^{-2\kappa_1}$. Hence, by Lemma A.10,

$$\|r\|_{L^2(\phi_d)} \leq \left(\frac{2B_0 e^{-2\kappa_1}}{n+1}\right)^{(n+1)/2} \leq e^{-\kappa_1 n} t \leq \frac{1}{2} e^{-\kappa_1 n} \|g\|_{L^2(\phi_d)}. \tag{50}$$

The last inequality follows from the Hölder's inequality $\|g\|_{L^1(\phi_d)} \leq \|g\|_{L^2(\phi_d)}$. We define $c_0 = c_0(\kappa_1, \kappa_2, M, d) := e^{-\kappa_1(A_1 d \vee B)}$ and conclude that

$$
\begin{aligned}
2t = \|g\|_{L^1(\phi_d)} &\geq \|q\|_{L^1(\phi_d)} - \|r\|_{L^1(\phi_d)} && (\because g = q + r) \\
&\geq c_{n,d} \|q\|_{L^2(\phi_d)} - \|r\|_{L^2(\phi_d)} && \text{(by (38))} \\
&\geq c_{n,d} \|g\|_{L^2(\phi_d)} - 2 \|r\|_{L^2(\phi_d)} && (\because c_{n,d} \leq 1) \\
&\geq 3e^{-\kappa_1 n} \|g\|_{L^2(\phi_d)} - e^{-\kappa_1 n} \|g\|_{L^2(\phi_d)} && \text{(by (45) and (50))} \\
&\geq 2 \exp\left( -\kappa_1 \left( A_1 d \vee B \vee \frac{2\kappa_2 \log(1/t)}{\log\left(\log(1/t) \vee e\right)} \right) \right) \|g\|_{L^2(\phi_d)} \\
&\geq 2 \left( c_0 \wedge t^{\alpha(t)} \right) \|g\|_{L^2(\phi_d)},
\end{aligned}
$$

where

$$
\alpha(t) = \frac{2\kappa_1 \kappa_2}{\log\left(\log(1/t) \vee e\right)}.
$$

Letting $C_0 := c_0^{-1}$ gives the desired result $\|g\|_{L^2(\phi_d)} \leq \left( C_0 \vee t^{-\alpha(t)} \right) t$. $\qquad\square$

### A.3. Dependency of the Constant

In this section, we discuss how the constant $C_0$ in the main Theorems 2.1 and 2.5 depends on the radius $M$ and dimension $d$. In short, $\log(C_0)$ has a polynomial order in $M^2 d$, and it is "nearly" linear in the regime where $\delta \to \infty$.

**Proposition A.11** (Dependency of $C_0$ on $M$ and $d$). *The constants $C_0 = C_0(\delta, M, d)$ in Theorems 2.1 and 2.5 coincide. Moreover, if we define $A_1 = A_1(\kappa_1)$ and $B = B(\kappa_1, \kappa_2, M^2 d)$ as in (40) and (48), respectively, then we can specify the constant as*

$$
\log(C_0) := \inf_{2\kappa_1 \kappa_2 = 2 + \delta} \kappa_1 (A_1 d \vee B),
$$

*where the infimum is taken with respect to $\kappa_1, \kappa_2 > 1$ such that $2\kappa_1 \kappa_2 = 2 + \delta$.*

*Proof.* The definition (40) of $A_1 = A_1(\kappa_1)$ reflects the assumption of Corollary A.9, which is required to meet the condition of Propositions A.7 and A.8 and to guarantee that $c_{n,d}$ defined in (38) is not less than $3e^{-\kappa_1 n}$, as demonstrated in the Corollary A.9. On the other hand, the definitions (47) and (48) of $B_0$ and $B$ reflect Lemma A.10, which is essential to control the tail norm $\|r\|_{L^2(\phi_d)}$ of $g = \frac{f_\pi - f_\eta}{\phi_d}$. We give more detailed discussion below. $\qquad\square$

The first observation is that once $\kappa_1 > 1$ is fixed, $A_1$ is merely a universal constant. This shows that $\log(C_0)$ must depend on the dimension $d$ at least linearly. In contrast, the behavior of $B_0$ and $B$ described in (47) and (48) is more intricate. It suffices to consider the regime where $2eM^2 d > 1$ because if the radius $M$ of support is too small, we can simply embed the support into a larger cube. Therefore, once $\kappa_1$ is fixed, we have $B_0 \asymp M^2 d$. If in (48) we are allowed to take $\kappa_2$ sufficiently large, then we would obtain $\log(C_0) \asymp B_0 \asymp M^2 d$. However, this cannot be achieved in the regime where $\delta > 0$ is fixed and $M^2 d$ is large. In such a situation, we have the following polynomial rate:

$$
\log(C_0) \asymp (M^2 d)^{\frac{\kappa_2 + 1}{\kappa_2 - 1}}.
$$

If $\delta > 0$ is taken sufficiently large, the polynomial order in $M^2 d$ may recover the limit $\frac{\kappa_2 + 1}{\kappa_2 - 1} \to 1$.

### A.4. Physical Interpretation: Quantum Harmonic Oscillator

In this section, we provide physical interpretation of the restricted-range inequality, Proposition A.7. A classical Hamiltonian of a particle in $\mathbb{R}^d$ is given by

$$
\mathcal{H}_{\mathrm{cl}} = \frac{1}{2} \|\xi\|_2^2 + V(x),
$$

where $\xi$ and $x$ are the momentum and position of the particle, respectively. The classical harmonic oscillator is defined by the potential energy $V(x) := \frac{1}{2}\|x\|_2^2$. The quantum-mechanical analog of the Hamiltonian is given by the following differential operator.

$$\mathcal{H} = -\frac{\hbar^2}{2}\nabla^2 + V : \psi \mapsto -\frac{\hbar^2}{2}\left(\frac{\partial^2}{\partial x_1^2} + \cdots + \frac{\partial^2}{\partial x_d^2}\right)\psi + \frac{1}{2}\left(x_1^2 + \cdots + x_d^2\right)\psi.$$

Here $\psi : \mathbb{R}^d \to \mathbb{R}$ is a wave function and $\hbar > 0$ is a constant closely related to the Planck constant, while we assume natural (mathematical) length and energy scales.

**Proposition A.12** (Isotropic quantum harmonic oscillator)**.** *For $\mathbf{k} \in \mathbb{N}_0^d$, define the Hermite function as*

$$\psi_{\mathbf{k}}(x) := \left(\frac{2}{\hbar}\right)^{d/4} h_{\mathbf{k}}\left(\sqrt{\frac{2}{\hbar}}x\right)\phi_d^{1/2}\left(\sqrt{\frac{2}{\hbar}}x\right).$$

*Then,*

1. *$\mathcal{H}$ is a self-adjoint operator.*

2. *(normalization) $\|\psi_{\mathbf{k}}\|_{L^2(\mathbb{R}^d)} = 1$.*

3. *(Schrödinger equation) $\mathcal{H}\psi_{\mathbf{k}} = E_{\mathbf{k}}\psi_{\mathbf{k}}$ where the eigenvalue is $E_{\mathbf{k}} = \frac{\hbar}{2}(2|\mathbf{k}| + d)$.*

4. *$\{\psi_{\mathbf{k}}\}$ consists entirely of eigenfunctions of $\mathcal{H}$.*

*Moreover, if we define the Mehler kernel $M(x, y; t) := \sum_{\mathbf{k}\in\mathbb{N}_0^d} e^{-tE_{\mathbf{k}}}\psi_{\mathbf{k}}(x)\psi_{\mathbf{k}}(y)$ for $t > 0$, then*

$$M(x, y; t) = (2\pi\hbar\sinh(\hbar t))^{-d/2}\exp\left(-\frac{\|x\|_2^2 + \|y\|_2^2}{2\hbar\tanh(\hbar t)} + \frac{\langle x, y\rangle_{L^2(\mathbb{R}^d)}}{\hbar\sinh(\hbar t)}\right).$$

*Proof.* See Lemma A.4. $\square$

*Remark* A.13. The eigenvalue $E_{\mathbf{k}}$ is the energy level of the state $\mathbf{k}$. A complex-analytical analog of Mehler kernel is the Feynman propagator, where $t > 0$ represents inverse temperature.

For the sake of the preceding proofs, we are only interested in the special case $\hbar = 2$, in which $\psi_{\mathbf{k}} = h_{\mathbf{k}}\phi_d^{1/2}$ and $E_{\mathbf{k}} = 2|\mathbf{k}| + d$. Recall that Corollary A.5 describes upper bounds of the quantity $K_n(x, x)\phi_d(x)$ involving the diagonal entries of Christoffel-Darboux kernel (26). The quantity can be rewritten as

$$K_n(x, x)\phi_d(x) = \sum_{E_{\mathbf{k}} \leq E_{n,d}} \psi_{\mathbf{k}}^2(x), \tag{51}$$

where $E_{n,d} = 2n + d$ as in (31). Thus, (51) represents the diagonal entries of low-energy spectral projector kernel and explains the spatial density of states (DOS). As such, local Weyl law states that, for every $x \in \mathbb{R}^d$, in the classical regime where $E_{n,d} \to \infty$, we have

$$\sum_{E_{\mathbf{k}} \leq E_{n,d}} \psi_{\mathbf{k}}^2(x) \to (4\pi)^{-d}\int_{\mathcal{H}_{\mathrm{cl}} \leq E_{n,d}} d\xi = (4\pi)^{-d}\omega_d\left(2E_{n,d} - \|x\|_2^2\right)^{d/2},$$

where $\omega_d$ is the volume of the $d$-dimensional unit (Euclidean) ball. Therefore, in the classically forbidden region where $\|x\|_2 > \sqrt{2E_{n,d}}$, i.e., the potential energy exceeds the mechanical energy, we expect the quantity (51) to converge to zero as $n \to \infty$. The tail bound (34) is the mathematically rigorous version of this intuition. Refer to Guillemin & Sternberg (2013) for further details.

# B. Proof of the Sharpness

This section completes the proof of our sharpness result by proving Lemma 3.2 and Corollary 3.3.

## B.1. Preliminaries: Chebyshev Polynomials and Lemmas

**Lemma B.1.** *Suppose $|\Delta_k| \leq 2b^k$ holds for all $k \in \mathbb{N}$. Then, there exists a universal $N \in \mathbb{N}$ such that*

$$n \geq N \vee (2.77)b^2 \implies \sum_{k=n+1}^{\infty} \frac{\Delta_k^2}{k!} \leq \left(\frac{eb^2}{n+1}\right)^{n+1}.$$

*Proof.* According to the Stirling's formula, there exists $N \in \mathbb{N}$, not depending on $b$, such that, if $n \geq N$,

$$\frac{\Delta_{n+\ell}^2}{(n+\ell)!} \leq \frac{4b^{2(n+\ell)}}{(n+\ell)!} \leq \left(1 - \frac{e}{2.77}\right)\left(\frac{eb^2}{n+\ell}\right)^{n+\ell}$$

holds for $\ell \geq 1$. If we assume further that $n \geq (2.77)b^2$, then

$$\sum_{\ell=1}^{\infty}\left(1 - \frac{e}{2.77}\right)\left(\frac{eb^2}{n+\ell}\right)^{n+\ell} \leq \sum_{\ell=1}^{\infty}\left(1 - \frac{e}{2.77}\right)\left(\frac{e}{2.77}\right)^{\ell-1}\left(\frac{eb^2}{n+1}\right)^{n+1} = \left(\frac{eb^2}{n+1}\right)^{n+1}.$$

$\square$

**Lemma B.2** (Chebyshev polynomials of the first kind). *Let $n \geq 11$ and $\theta_j = \cos\left(\frac{2j+1}{2n+2}\pi\right), j = 0, \ldots, n$ be the zeros of Chebyshev polynomial of the first kind, $T_{n+1}(x)$, with degree $n+1$. Then,*

1. $|T_{n+1}(t\sqrt{-1})| = \left\{(t + \sqrt{t^2+1})^{n+1} + (t - \sqrt{t^2+1})^{n+1}\right\}/2$ *holds for $t > 0$.*

2. $z_n = \sqrt{-\frac{n}{2.77}} \in \mathbb{C}$ *satisfies* $\frac{1}{2^n|z_n|^{n+1}}|T_{n+1}(z_n)| < 2$.

3. $\left\|V_{n+1}^{-1}\right\|_{\infty} \leq \frac{(1+\sqrt{2})^{n+1}}{n+1}$, *where*

$$V_{n+1} = \begin{bmatrix} 1 & \cdots & 1 \\ \vdots & \ddots & \vdots \\ \theta_0^n & \cdots & \theta_n^n \end{bmatrix}$$

   *is the $(n+1) \times (n+1)$ Vandermonde matrix involving $\theta_0, \ldots, \theta_n$.*

*Proof.* First, applying de Moivre's formula to the definition (8) gives

$$T_{n+1}(x) = \frac{1}{2}\left(\zeta^{n+1} + \zeta^{-(n+1)}\right),$$

where $x \in \mathbb{C}$ and $\zeta = x \pm \sqrt{x^2 - 1}$. (No matter which branch is chosen for the square root, the two summands are reciprocal to each other.) Second, if $z_n = \sqrt{-\frac{n}{2.77}}$, then

$$\frac{1}{2^n|z_n|^{n+1}}|T_{n+1}(z_n)| = \left(\frac{1 + \sqrt{1 + \frac{2.77}{n}}}{2}\right)^{n+1} + \left(\frac{1 - \sqrt{1 + \frac{2.77}{n}}}{2}\right)^{n+1}$$

$$\rightarrow \exp\left(\frac{2.77}{4}\right) < 2,$$

as $n \rightarrow \infty$. (A more careful computation shows $n \geq 11$ is sufficient.) Finally, according to Example 6.2 of Gautschi (1974), we have

$$\left\|V_{n+1}^{-1}\right\|_{\infty} \leq \frac{3^{3/4}}{2(n+1)}|T_{n+1}(\sqrt{-1})| \leq \frac{(1+\sqrt{2})^{n+1}}{n+1}.$$

$\square$

**Lemma B.3.** *Let $n$ be a positive odd integer. Then,*

$$\max\left\{\frac{(n/2.77)^\ell}{(2\ell)!!} : \ell = 0, \ldots, \frac{n-1}{2}\right\} \leq \exp\left(\frac{n}{5.54}\right),$$

*where $(2\ell)!!$ denotes a double factorial.*

*Proof.* For $\ell \geq 1$, we have $(2\ell)!! = 2^\ell \ell!$ and

$$\frac{(n/2.77)^\ell}{2^\ell \ell!} \leq \left(\frac{en}{5.54\ell}\right)^\ell \leq \exp\left(\frac{n}{5.54}\right).$$

The first inequality holds from the Stirling's formula and the second one is given by optimizing with respect to $\ell$ over positive reals. The optimal value is attained at $\ell = n/(5.54)$. $\qquad\square$

## B.2. Proofs

We now proceed to prove Lemma 3.2 and Corollary 3.3.

*Proof of Lemma 3.2.* We solve the following linear system:

$$\begin{bmatrix} 1 & & 0 \\ & \ddots & \\ 0 & & a^n \end{bmatrix} \begin{bmatrix} 1 & \cdots & 1 \\ \vdots & \ddots & \vdots \\ \theta_0^n & \cdots & \theta_n^n \end{bmatrix} \begin{bmatrix} w_0 \\ \vdots \\ w_n \end{bmatrix} = \begin{bmatrix} \Delta_0 \\ \vdots \\ \Delta_n \end{bmatrix}.$$

By the third statement of Lemma B.2, we have $|w_j| \leq \left\|V_{n+1}^{-1}\right\|_\infty a^{-n}\Delta_n \leq \frac{1}{n+1}$ for all $j$. Indeed, $\pi_n^{(0)}$ and $\eta_n^{(0)}$ are valid probability measures supported on $[-M, M]$ since $\sum_{j=0}^n w_j = \Delta_0 = 0$. We also have

$$\Delta_k = \sum_{j=0}^n w_j(a\theta_j)^k = \int \theta^k d(\pi_n^{(0)} - \eta_n^{(0)})(\theta),$$

for $k = 0, 1, \ldots, n$. Lemma B.3 verifies that (10) holds for all $0 \leq k \leq n$. We will now use mathematical induction to show that, in fact, (10) holds for all $k \geq 0$. Let $K \geq n$ and assume the induction hypothesis (10) to be true for all $k \leq K$. Recall that

$$T_{n+1}(x) = 2^n(x^{n+1} - \sigma_2 x^{n-1} + \sigma_4 x^{n-3} - \cdots + (-1)^{(n+1)/2}\sigma_{n+1}),$$

where $\sigma_m$ denotes the $m$-th elementary symmetric function of the zeros $\theta_0, \ldots, \theta_n$. Since $T_{n+1}(\theta_j) = 0$,

$$(a\theta_j)^{K+1} = \sigma_2 a^2 (a\theta_j)^{K-1} - \sigma_4 a^4 (a\theta_j)^{K-3} + \cdots + (-1)^{(n-1)/2}\sigma_{n+1}a^{n+1}(a\theta_j)^{K-n},$$

$$\begin{aligned}
|\Delta_{K+1}| &= \left|\sum_{j=0}^n w_j(a\theta_j)^{K+1}\right| \\
&\leq \sigma_2 a^2|\Delta_{K-1}| + \sigma_4 a^4|\Delta_{K-3}| + \cdots + \sigma_{n+1}a^{n+1}|\Delta_{K-n}| \\
&\leq \left\{a(\sqrt{2}-1)\right\}^{n+1}\exp\left(\frac{n}{5.54}\right)b^{K+1-n}\left(\sigma_2(a/b)^2 + \sigma_4(a/b)^4 + \cdots\right) \\
&= \left\{a(\sqrt{2}-1)\right\}^{n+1}\exp\left(\frac{n}{5.54}\right)b^{K+1-n}\left(\frac{a^{n+1}}{2^n b^{n+1}}\left|T_{n+1}\left(\frac{b}{a}\sqrt{-1}\right)\right| - 1\right) \\
&\leq \left\{a(\sqrt{2}-1)\right\}^{n+1}\exp\left(\frac{n}{5.54}\right)b^{K+1-n}.
\end{aligned}$$

The last inequality follows from the second statement of Lemma B.2. We have shown that the induction hypothesis (10) is also true for $k = K + 1$. Thus, (10) is true for all $k \geq 0$. Now, we proceed to prove the very last statement. In view of

Lemma B.1, there exists $N \in \mathbb{N}$, not depending on $a$ or $b$, such that if $n \geq N$, then

$$\|r_n\|_{L^2(\phi)} \leq \left\{a(\sqrt{2}-1)\right\}^{n+1} \exp\left(\frac{n}{5.54}\right) b^{-n} \left(\frac{eb^2}{n+1}\right)^{(n+1)/2}$$

$$\leq \left\{a(\sqrt{2}-1)\right\}^{n+1} \exp\left(\frac{n}{5.54}\right) \sqrt{\frac{n}{2.77}} \left(\frac{e}{n+1}\right)^{(n+1)/2}$$

$$\leq \left\{a(\sqrt{2}-1)\right\}^{n+1} \exp\left(\frac{n}{5.54}\right) \left(\frac{e}{n}\right)^{n/2}.$$

Lastly, observing that

$$q_n(x) = \sum_{\ell=0}^{(n-1)/2} \left\{a(\sqrt{2}-1)\right\}^{n+1} \frac{h_{n-2\ell}(x)}{(2\ell)!!\sqrt{(n-2\ell)!}}$$

$$= \left\{a(\sqrt{2}-1)\right\}^{n+1} \frac{x^n}{n!}$$

gives the following explicit formulas for $L^1(\phi)$ and $L^2(\phi)$ norms of $q_n$.

$$\|q_n\|_{L^1(\phi)} = \left\{a(\sqrt{2}-1)\right\}^{n+1} \frac{2^{n/2}\pi^{-1/2}\Gamma\left(\frac{n+1}{2}\right)}{n!}$$

$$= \left\{a(\sqrt{2}-1)\right\}^{n+1} (\pi n)^{-1/2} \left(\frac{e}{n}\right)^{n/2} \left(1 + O\left(\frac{1}{n}\right)\right),$$

$$\|q_n\|_{L^2(\phi)} = \left\{a(\sqrt{2}-1)\right\}^{n+1} \frac{2^{n/2}\pi^{-1/4}\Gamma^{1/2}\left(n+\frac{1}{2}\right)}{n!}$$

$$= \left\{a(\sqrt{2}-1)\right\}^{n+1} (\pi n)^{-1/2} \left(\frac{e}{n}\right)^{n/2} 2^{\frac{n}{2}-\frac{1}{4}} \left(1 + O\left(\frac{1}{n}\right)\right).$$

Comparing these asymptotics shows (11) and (12). In particular, both $\|q_n\|_{L^1(\phi)}$ and $\|q_n\|_{L^2(\phi)}$ decay in a hyper-exponential rate of $\exp(-n\log n/2)$, and the tail norm $\|r_n\|_{L^2(\phi)}$ cannot deviate from $\|q_n\|_{L^1(\phi)}$ or $\|q_n\|_{L^2(\phi)}$ faster than an exponential rate in $n$. We have (13) in conclusion. $\qquad\square$

*Proof of Corollary 3.3.* The equality for the total variation distance is straightforward. In view of Lemma 3.2, let $n$ be a large enough odd integer. By construction, we have

$$f_{\pi_n^{(1)}}(x) = (1-\lambda_n)\phi(x) + \sum_{j=0}^{n} \left(\frac{\lambda_n}{n+1} + \lambda_n w_j\right)\phi(x - a\theta_j),$$

$$f_{\eta_n^{(1)}}(x) = (1-\lambda_n)\phi(x) + \sum_{j=0}^{n} \frac{\lambda_n}{n+1}\phi(x - a\theta_j).$$

Recall from the lemma that $|\theta_j| \leq 1$ for all $j$ and that $0 < a \leq 1$. Also, recall the definition (14) of $R_n$ and $\lambda_n$. Observe for all $x \in [-R_n, R_n]$ and $j$ that

$$\frac{\phi(x - a\theta_j)}{\phi(x)} = \exp\left(a\theta_j x - \frac{1}{2}a^2\theta_j^2\right) \leq \exp(|a\theta_j|R_n) \leq \exp(R_n)$$

and that

$$f_{\eta_n^{(1)}}(x) \leq (1 - \lambda_n + \lambda_n\exp(R_n))\,\phi(x) \leq 2\phi(x). \tag{52}$$

Lastly, recall the definition (31) of $E_{n,d}$. Note that $E_{n,1} = 2n+1$ and that $R_n = \sqrt{8n+4} = \sqrt{2\kappa E_{n,1}}$ holds for $\kappa = 2$.

Therefore, we have

$$
\begin{aligned}
\frac{2}{\lambda_n^2} \chi^2 \left( f_{\pi_n^{(1)}} \| f_{\eta_n^{(1)}} \right) &\geq \frac{2}{\lambda_n^2} \int_{-R_n}^{R_n} \frac{\left( f_{\pi_n^{(1)}} - f_{\eta_n^{(1)}} \right)^2}{f_{\eta_n^{(1)}}} \\
&\geq \int_{-R_n}^{R_n} \frac{\left( f_{\pi_n^{(0)}} - f_{\eta_n^{(0)}} \right)^2}{\phi} && \text{(by (52))} \\
&= \| q_n + r_n \|_{L^2([-R_n, R_n], \phi)}^2 \\
&\geq \frac{1}{2} \| q_n \|_{L^2([-R_n, R_n], \phi)}^2 - \| r_n \|_{L^2(\phi)}^2 && (\because 2(a^2 + b^2) \geq (a-b)^2) \\
&\geq \frac{1}{4} \| q_n \|_{L^2(\phi)}^2 - \| r_n \|_{L^2(\phi)}^2 && \text{(by Proposition A.7 with } \kappa = 2\text{)} \\
&\geq \frac{1}{8} \| q_n \|_{L^2(\phi)}^2, && \text{(by inequality (12))}
\end{aligned}
$$

provided that $n$ is large enough. $\qquad\square$

## C. Proof of the Applications

In this section, we prove Theorem 4.3, Proposition 4.4, Theorems 4.5, 4.6, and 4.7.

### C.1. Preliminaries: Yatracos' Construction and Lemmas

We first recall the application of Yatracos' scheme idea (Yatracos, 1985) for robust density estimation in total variation.

Consider an $\eta$-covering $\{Q_1, \ldots, Q_N\}$ of $\mathcal{P}_{M,d}$ in total variation. Then, we define the Yatracos' class $\mathcal{A}$ by

$$
\begin{aligned}
\mathcal{A} &:= \{ A_{ij} : i \neq j \in [N] \}, \\
A_{ij} &:= \left\{ x : \frac{dQ_i}{d(Q_i + Q_j)}(x) \geq \frac{dQ_j}{d(Q_i + Q_j)}(x) \right\},
\end{aligned}
$$

so that $|\mathcal{A}| \leq N^2$. Given the class $\mathcal{A}$, we define a pseudo-distance $\mathrm{dist}$ as follows.

$$
\mathrm{dist}(P_1, P_2) := \sup_{A \in \mathcal{A}} |P_1(A) - P_2(A)|.
$$

Then, $\mathrm{dist}$ satisfies triangular inequality. Moreover, it approximates the total variation on $\mathcal{P}_{M,d}$, in the sense that

$$
\begin{aligned}
\mathrm{dist}(Q_i, Q_j) &= \mathrm{TV}(Q_i, Q_j), \\
\mathrm{dist}(P_1, P_2) &\leq \mathrm{TV}(P_1, P_2) \leq \mathrm{dist}(P_1, P_2) + 4\eta, && \forall P_1, P_2 \in \mathcal{P}_{M,d}.
\end{aligned}
$$

Given i.i.d. observations $X_1, \ldots, X_n$ as in (20), we define the Yatracos' estimator $\widehat{P}$ by

$$
\widehat{P} := \operatorname*{argmin}_{P' \in \mathcal{P}_{M,d}} \mathrm{dist}\left( P', \widehat{P}_n \right), \tag{53}
$$

where $\widehat{P}_n := \frac{1}{n} \sum_{i=1}^{n} \delta_{X_i}$ is the empirical distribution. Note that the Yatracos' scheme works even if $P = (1-\epsilon)P_{f_\pi} + \epsilon Q$ is outside $\mathcal{P}_{M,d}$. In particular, we have

$$
\mathrm{TV}\left( P, \widehat{P} \right) \leq 3 \inf_{P' \in \mathcal{P}_{M,d}} \mathrm{TV}\left( P, P' \right) + 3\eta + 2\,\mathrm{dist}\left( P, \widehat{P}_n \right). \tag{54}
$$

See Section 32.3 of Polyanskiy & Wu (2025) for recent review on the Yatracos' estimator. As a consequence, we can derive the minimax upper bound in Proposition 4.4, noting that $\log N \lesssim \log^{d+1}(1/\eta)$ holds from Lemma C.1. It only remains to choose appropriate $\eta$ for (54). See Appendix C.2 for the details.

**Lemma C.1** (TV entropy bound in $d$ dimension). *Recall the definition of covering number from Definition 4.1. We have*

$$\log N_{\mathrm{TV}}(\mathcal{P}_{M,d}, \eta) \lesssim \log^{d+1}\left(\frac{1}{\eta}\right).$$

*Proof.* For the one-dimensional case ($d = 1$), the entropy bound is due to Ghosal & Van der Vaart (2001). Recent works extended this result to arbitrary dimensions (Saha & Guntuboyina, 2020; Ma et al., 2025). Let $\mathcal{P}_m$ be the collection of $m$-atomic Gaussian mixtures in $\mathcal{P}_{M,d}$ and define

$$m^\star := \inf\left\{m \in \mathbb{N} : \sup_{P \in \mathcal{P}_{M,d}} \inf_{P_m \in \mathcal{P}_m} \mathrm{TV}(P, P_m) \leq \frac{\eta}{2}\right\}.$$

Then, Proposition 5 of Ma et al. (2025) shows $m^\star \lesssim \log^d(1/\eta)$. On the other hand, parametric entropy bound on finite mixtures shows

$$\log N_{\mathrm{TV}}\left(\mathcal{P}_{m^\star}, \frac{\eta}{2}\right) \lesssim m^\star d \log\left(\frac{1}{\eta}\right).$$

Combining these results with triangular inequality concludes the proof. $\qquad\square$

**Lemma C.2** (Chen et al. (2018)). *Suppose $P_1$ and $P_2$ are probability measures such that $\mathrm{TV}(P_1, P_2) \leq \frac{\epsilon}{1-\epsilon}$. Then, there exist two probability measures $Q_1$ and $Q_2$ such that $(1-\epsilon)P_1 + \epsilon Q_1 = (1-\epsilon)P_2 + \epsilon Q_2$.*

### C.2. Proofs

We proceed to prove Theorem 4.3, Proposition 4.4, Theorems 4.5, 4.6, and 4.7 in this section.

*Proof of Theorem 4.3.* First, for one estimator $\widehat{P}$, suppose $\widetilde{P}$ is the projection of $\widehat{P}$ onto $\mathcal{P}$ under TV distance. Then, for every $P \in \mathcal{P}$, we have

$$\mathrm{TV}\left(P, \widetilde{P}\right) \leq \mathrm{TV}\left(P, \widehat{P}\right) + \mathrm{TV}\left(\widehat{P}, \widetilde{P}\right) \leq 2\mathrm{TV}\left(P, \widehat{P}\right).$$

This allows $\widehat{P}$ to be restricted to $\mathcal{P}$ up to universal constants.

Second, the upper bound follows immediately from the inequality (1) and Proposition 4.2.

Third, applying Corollary 2.4 gives

$$\mathbb{P}\left[\mathrm{TV}\left(P, \widehat{P}\right) \geq \mathcal{J}^{-1}\left(\frac{\epsilon_n}{4}\right)\right] \geq \mathbb{P}\left[H\left(P, \widehat{P}\right) \geq \frac{\epsilon_n}{4}\right] \geq \frac{1}{2}, \tag{55}$$

where we define $\alpha(t)$ as in (5) and $\mathcal{J}(t)$ as

$$\mathcal{J}(t) := C_0 t \vee t^{1-\alpha(t)}, \tag{56}$$

for $t > 0$. Note that the inverse $\mathcal{J}^{-1}$ is well-defined in the regime where $n \to \infty$ as $\mathcal{J}$ is strictly increasing in $(0, t_0)$ for some $t_0 > 0$. The last inequality in (55) is due to Fano's inequality used in the proof of Corollary 11 of Jia et al. (2023). We conclude that

$$\inf_{\widehat{P} \in \mathcal{P}} \sup_{P \in \mathcal{P}} \mathbb{E}_P\left[\mathrm{TV}^2\left(P, \widehat{P}\right)\right] \gtrsim \left(\mathcal{J}^{-1}\left(\frac{\epsilon_n}{4}\right)\right)^2$$

$$\gtrsim \epsilon_n^{2\left(1 + \frac{2+\delta}{\log(\log(1/\epsilon_n)\vee e)}\right)}.$$

$\qquad\square$

*Proof of Proposition 4.4.* The standard Yatracos' construction (53) leads to a proper estimator $\widehat{P} \in \mathcal{P}_{M,d}$. We denote by $\widehat{f}$ the density of $\widehat{P}$. Observe that

$$\inf_{P' \in \mathcal{P}_{M,d}} \mathrm{TV}\left(P, P'\right) \leq \mathrm{TV}\left(P, P_{f_\pi}\right) \leq \epsilon.$$

Hence, by (54), $\widehat{f}$ satisfies

$$\mathrm{TV}\left(f_\pi, \widehat{f}\right) \leq \epsilon + \mathrm{TV}\left(P, \widehat{P}\right) \leq 4\epsilon + 3\eta + 2\,\mathrm{dist}\left(P, \widehat{P}_n\right). \tag{57}$$

Applying the Hoeffding bound and union bound, we have

$$\mathbb{P}\left(\mathrm{dist}\left(P, \widehat{P}_n\right) \geq s\right) \leq 1 \wedge 2|\mathcal{A}| \exp\left(-\frac{ns^2}{2}\right),$$

$$\mathbb{E}_P\left(\mathrm{dist}^2\left(P, \widehat{P}_n\right)\right) \leq \frac{2\left(1 + \log\left(2|\mathcal{A}|\right)\right)}{n}.$$

Lemma C.1 implies

$$\log|\mathcal{A}| \leq 2\log N_{\mathrm{TV}}(\mathcal{P}_{M,d}, \eta) \lesssim \log^{d+1}\left(1/\eta\right).$$

Accordingly, we choose optimal $\eta \asymp \log^{d/2}(n)/\sqrt{n}$ to conclude the proof. $\qquad\square$

*Proof of Theorem 4.5.* Let $\widehat{f}$ be the proper estimator from the proof of Proposition 4.4. We first show that the function $G(t) := t^{1-\alpha(t)}$ is strictly increasing and concave on an open interval $t < t_0$, where $\alpha(t) = \frac{c}{\log\log(1/t)}$, $c = 2 + \delta$, and $t_0 < e^{-e}$ is a constant depending only on $\delta$. To this end, we take derivatives:

$$G'(t) = \frac{G(t)}{t}\left(1 - \frac{c}{\log\log(1/t)} + \frac{c}{\log^2\log(1/t)}\right), \qquad G''(t) = -\frac{G(t)}{t^2}\left(\frac{c}{\log\log(1/t)} - \frac{c^2 + c + o(1)}{\log^2\log(1/t)}\right),$$

verifying that $G'(t) > 0$ and $G''(t) < 0$ hold for all $t < t_0$. We also note that $\lim_{t \searrow 0} G(t) = 0$. Therefore, given that $C_0$ is not less than $t_0^{-\alpha(t_0)}$, a constant depending only on $\delta$, there exists $0 < t_1 \leq t_0$ such that

$$\mathcal{J}(t) = C_0 t \vee G(t) = \begin{cases} C_0 t, & t \geq t_1, \\ G(t), & t < t_1, \end{cases}$$

where we define $\mathcal{J}(\cdot)$ as in (56). Using the concavity of $G$, we obtain for all $s, t > 0$ that

$$\begin{aligned} \mathcal{J}(s+t) &= C_0(s+t) \vee G(s+t) \\ &\leq C_0(s+t) \vee (G(s) + G(t)) \\ &\leq (C_0 s \vee G(s)) + (C_0 t \vee G(t)) \\ &= \mathcal{J}(s) + \mathcal{J}(t). \end{aligned}$$

The second line is due to the fact that $C_0(s+t) < G(s+t)$ implies $G(s+t) \leq G(s) + G(t)$, and the third line is due to the general inequality: $(a+c) \vee (b+d) \leq (a \vee b) + (c \vee d)$. Thus, applying Corollary 2.4 to (57) gives

$$H\left(f_\pi, \widehat{f}\right) \leq 4\mathcal{J}(\epsilon) + 3\mathcal{J}(\eta) + 2\mathcal{J}\left(\mathrm{dist}\left(P, \widehat{P}_n\right)\right).$$

Hence, by the Cauchy-Schwarz inequality,

$$H^2\left(f_\pi, \widehat{f}\right) \leq \left(4^2 + 3^2 + 2^2\right)\left[\mathcal{J}^2(\epsilon) + \mathcal{J}^2(\eta) + \mathcal{J}^2\left(\mathrm{dist}\left(P, \widehat{P}_n\right)\right)\right].$$

Taking expectations on both sides yields the desired bound (21) by choosing the same $\eta$ as in the proof of Proposition 4.4. $\quad\square$

*Proof of Theorem 4.6.* The minimax lower bound in $\epsilon$ can be obtained from standard two-point method. Our sharpness result, Theorem 3.1, shows that there exist two "one-dimensional" probability measures $\pi^\star$ and $\eta^\star$, supported on the bounded interval $[-M, M]$, such that $\mathrm{TV}(f_{\pi^\star}, f_{\eta^\star}) \leq \epsilon \leq \frac{\epsilon}{1-\epsilon}$ and that

$$H(f_{\pi^\star}, f_{\eta^\star}) \gtrsim \epsilon^{\left(1 - \frac{0.33}{\log(\log(1/\epsilon) \vee e)}\right)}.$$

Note that we can also construct $d$-dimensional probability measures $\pi$ and $\eta$ with the same property because $\mathrm{TV}(f_\pi, f_\eta) = \mathrm{TV}(f_{\pi^\star}, f_{\eta^\star})$ and $H(f_\pi, f_\eta) = H(f_{\pi^\star}, f_{\eta^\star})$ for

$$\pi = \pi^\star \otimes \delta_0^{\otimes(d-1)} = \pi^\star \otimes \delta_0 \otimes \cdots \otimes \delta_0,$$
$$\eta = \eta^\star \otimes \delta_0^{\otimes(d-1)} = \eta^\star \otimes \delta_0 \otimes \cdots \otimes \delta_0,$$

where $\delta_0$ denotes the point mass at zero and $\otimes$ the product measure. Thus, it follows from Lemma C.2 and the same two-point argument in Chen et al. (2018) that

$$\inf_{\widehat{f}} \sup_{\pi, Q} \mathbb{E}\left[H^2\left(f_\pi, \widehat{f}\right)\right] \gtrsim \epsilon^{2\left(1 - \frac{0.33}{\log(\log(1/\epsilon) \vee e)}\right)}.$$

$\square$

*Proof of Theorem 4.7.* This proof crucially relies on the proof of Theorem 3.5 of Saha & Guntuboyina (2020). Our proof, however, differs from theirs in the choice of $\rho$: they take $\rho = (2\pi)^{-d/2} n^{-1}$, whereas we use

$$\rho = (2\pi)^{-d/2}\left(\mathcal{E}^2(\epsilon, n) \wedge e^{-2}\right),$$

where we define $\mathcal{E}^2(\epsilon, n)$ as in (22).

Recall that the oracle Bayes estimator $\widehat{\theta}^\star(\cdot)$ is given by (23), and consider the following decomposition:

$$\mathbb{E}_{X \sim f_\pi}\left\|\widehat{\theta}_\rho(X) - \widehat{\theta}^\star(X)\right\|^2 \leq 2\mathbb{E}_{X \sim f_\pi}\left\|\widehat{\theta}_\rho(X) - \widehat{\theta}_\rho^\star(X)\right\|^2 + 2\mathbb{E}_{X \sim f_\pi}\left\|\widehat{\theta}_\rho^\star(X) - \widehat{\theta}^\star(X)\right\|^2, \tag{58}$$

where we define

$$\widehat{\theta}_\rho^\star(X) := X + \frac{\nabla f_\pi(X)}{f_\pi(X) \vee \rho}.$$

The first term of (58) is bounded from above as follows using Theorem E.1 of Saha & Guntuboyina (2020) together with the discretization argument from the proof of Theorem 3.5 therein.

$$\mathbb{E}_{X \sim f_\pi}\left\|\widehat{\theta}_\rho(X) - \widehat{\theta}_\rho^\star(X)\right\|^2 = \int \left\|\frac{\nabla \widehat{f}(x)}{\widehat{f}(x) \vee \rho} - \frac{\nabla f_\pi(x)}{f_\pi(x) \vee \rho}\right\|^2 f_\pi(x)\, dx$$
$$\lesssim H^2\left(f_\pi, \widehat{f}\right)\left(\log \frac{1}{H\left(f_\pi, \widehat{f}\right)} \vee \log^3\left(\frac{1}{\mathcal{E}(\epsilon, n)} \vee e\right)\right).$$

For the second term of (58), we have

$$\mathbb{E}_{X \sim f_\pi}\left\|\widehat{\theta}_\rho^\star(X) - \widehat{\theta}^\star(X)\right\|^2 = \int \left\|\frac{\nabla f_\pi(x)}{f_\pi(x) \vee \rho} - \frac{\nabla f_\pi(x)}{f_\pi(x)}\right\|^2 f_\pi(x)\, dx$$
$$= \int \left(1 - \frac{f_\pi(x)}{f_\pi(x) \vee \rho}\right)^2 \frac{\|\nabla f_\pi(x)\|^2}{f_\pi(x)}\, dx$$
$$\lesssim \mathcal{E}^2(\epsilon, n) \log^d\left(\frac{1}{\mathcal{E}(\epsilon, n)} \vee e\right).$$

The last inequality follows from Lemma 4.3 of Saha & Guntuboyina (2020).

Recall from our Theorem 4.5 that

$$\mathbb{E}\left[H^2\left(f_\pi, \widehat{f}\right)\right] \lesssim \mathcal{E}^2(\epsilon, n).$$

For brevity, write $H := H\left(f_\pi, \widehat{f}\right)$ and $\mathcal{E} := \mathcal{E}(\epsilon, n)$ for the remainder of the proof. Observe that the function $H \mapsto H^2 \log \frac{1}{H}$ is bounded from above, and it is strictly increasing for $H \le e^{-1}$. Thus,

$$
\begin{aligned}
&\mathbb{E}\left[H^2 \log \frac{1}{H}\right] \\
&= \mathbb{E}\left[H^2 \log \frac{1}{H}\mathbf{1}\{H \le \mathcal{E} \le e^{-1}\}\right] + \mathbb{E}\left[H^2 \log \frac{1}{H}\mathbf{1}\{H \le \mathcal{E}\}\mathbf{1}\{\mathcal{E} > e^{-1}\}\right] + \mathbb{E}\left[H^2 \log \frac{1}{H}\mathbf{1}\{H > \mathcal{E}\}\right] \\
&\le \mathcal{E}^2 \log\left(\frac{1}{\mathcal{E}} \vee e\right) + \mathbb{E}\left[H^2 \log \frac{1}{H}\mathbf{1}\{\mathcal{E} > e^{-1}\}\right] + \mathbb{E}\left[H^2\right] \mathbb{P}\left[H^2 > \mathcal{E}^2\right] \log\left(\frac{1}{\mathcal{E}} \vee e\right) \\
&\lesssim \mathcal{E}^2 \log\left(\frac{1}{\mathcal{E}} \vee e\right). \hspace{5cm} \text{(by Markov inequality)}
\end{aligned}
$$

Taking all into account, we conclude that

$$\mathbb{E}\left[\mathbb{E}_{X \sim f_\pi}\left\|\widehat{\theta}_\rho(X) - \widehat{\theta}^\star(X)\right\|^2\right] \lesssim \mathcal{E}^2 \log^{3 \vee d}\left(\frac{1}{\mathcal{E}} \vee e\right) \lesssim \epsilon^{2\left(1 - \frac{2 + 2\delta}{\log(\log(1/\epsilon) \vee e)}\right)} + n^{-(1 - o_d(1))}.$$

Note that the extra logarithmic factors are absorbed into the slack parameter $\delta > 0$ and $o_d(1)$, respectively. Since the choice of $\delta > 0$ is arbitrary, replace $\delta$ with $\delta/2$ to prove the bound (25). $\qquad\square$

