# OpenReview forum: "Sharp Inequalities between Total Variation and Hellinger Distances for Gaussian Mixtures"
_ICML.cc/2026/Conference — ICML 2026 spotlight_

### Official Review · Reviewer_YaoY · 2026-03-07

**Soundness:** 3
**Presentation:** 3
**Significance:** 4
**Originality:** 3
**Overall Recommendation:** 5
**Confidence:** 2

**Summary:**

This paper looks at the Hellinger distance between two mixture of Gaussians and asks whether or not the Hellinger distance between two GMMs is within a constant factor of the TV distance. Here, the specifically focus on location mixtures, i.e. the covariance matrix is just the identity matrix. It turns out that KL and Hellinger are the same up to constant factors.

In this paper, the authors prove that Hellinger and TV are *not* within constant factor of each other. Specifically, they prove that $H \leq TV^{1 - o(1)}$ where the $o(1)$ is something like $1 / \log \log TV$. They also prove that this inequality is sharp.

The proofs themselves are quite technical.

**Compliance With Llm Reviewing Policy:**

Affirmed.

**Final Justification:**

I think this is a good paper from a theoretical point of view so I support acceptance. However, note that I am not able to verify the proof. If the proof is correct then I think this is a good paper to publish and should be accepted. These sort of inequalities tend to be useful at some point for other applications as well.

**Key Questions For Authors:**

N/A

**Limitations:**

yes

**Strengths And Weaknesses:**

**Strengths.** Overall, I think this is a good paper. It's a fundamental question in statistics which the authors resolve. The techniques seem to be intricate and highly technical which I did not check carefully. Such a result will definitely be of interest to researchers that do more theoretical statistics.

**Weaknesses.** The paper looks at mixtures of location Gaussians. This is a bit restrictive but given that no prior results are known I don't think is a dealbreaker.
In terms of presentation, it is highly technical. Of course, it would be great if the presentation can be simplified but I don't think this is a big deal. It would be good if this is eventually submitted to a journal so that it can be checked more carefully.

Overall, given that this paper proves a fairly fundamental result in statistics, I would recommend accepting the paper.

---

> ### Author Rebuttal · Authors · 2026-03-31
>
> We sincerely thank the reviewer for the positive evaluation, for recognizing the statistical importance of our work, and for highlighting the weaknesses.
>
> We acknowledge that the proofs and presentation are technical. To address these concerns and make the paper more accessible, we plan to add more high-level intuition and prose discussions to the main text. Specifically, we will explicitly outline the technical roadblocks, the high-level proof architecture, and the role of our main mathematical tools (such as Chebyshev nodes and Nikolskii-type inequalities).

---

> > ### Author Rebuttal · Reviewer_YaoY · 2026-04-01
> >
> > No questions; just a comment about readability and accessibility to non-experts.
> >
> > Will keep my score.

---

### Official Review · Reviewer_zMsC · 2026-03-12

**Soundness:** 2
**Presentation:** 3
**Significance:** 3
**Originality:** 3
**Overall Recommendation:** 5
**Confidence:** 4

**Summary:**

The paper studies the relation between the TV distance and Hellinger distance between bounded-support Guassian mixtures. First, the authors prove an upper bound on the Hellinger distance in terms of TV. Then, explicit constructions are provided to prove the sharpness (upto constants) of the derived upper bound. Furthermore, these results are later used to characterize minimax rates in learning gaussian mixtures (under total variation) and robust density estimation for gaussian mixture under a huber contamination model.

**Compliance With Llm Reviewing Policy:**

Affirmed.

**Final Justification:**

My concerns were addressed. I will maintain my score.

**Key Questions For Authors:**

1. In the upper and lower bounds proved for Hellinger distance in Corollary 2.4 and Theorem 3.1, respectively, the gap between the constants 2+\delta and 0.33, is it just a proof slack, or is there a more fundamental obstruction to matching the constants more closely?

2. The paper assumes across all the results that the mixtures are supported on [-M, M]^d. While the proof sketch uses this assumption in the moment growth bound, how essential is this assumption? Can subgaussian/subexponential tails yield a similar result?

3. The proof of Theorem 3.1 appears to rely on Corollaries 3.3 and 3.4 (Lemma 3.2, for instance, n>=11), which are stated for large enough numbers, i.e., n >= N_0. Could authors clarify if the theorem should read ' for all n sufficiently large'or the small n case is handled separately?

4. For the construction in Theorem 3.1 for sharpness, I would like to know how sensitive the analysis is to the choice of Chebyshev zeros for inverse Vandermonde matrix stability. Would another stable design fail badly, or is it a convenient choice?

5. Potential typos: Line 183, Column 2: then modify --->then modifies

**Limitations:**

yes

**Strengths And Weaknesses:**

Strengths:

1. The main results (Theorem 2.1 and Corollary 2.4) are technically sound and resolve a previously stated open question by Jia et al.

2. The proof techniques are non-trivial and interesting, especially the Hermite polynomial-based framework and the multivariate analysis beyond 1D.

3. The authors present the applicability of their results in context to learning rates in terms of TV, and also study robust density estimation.

4. The paper is well-written and comprehensible to a reader. The results are presented clearly, with all notation and abbreviations defined properly.

Weaknesses:

1. In my opinion, the contribution is primarily structural/theoretical with limited or negligible algorithmic or empirical analysis.

2. The whole analysis in the paper revolves around 2 major assumptions: i. bounded Gaussian mixtures, ii. isotropic and fixed covariance mixtures.

---

> ### Author Rebuttal · Authors · 2026-03-31
>
> We sincerely thank the reviewer for recognizing the importance of our main results and proof techniques, and for providing a positive evaluation and insightful feedback. We agree that our contribution is primarily theoretical. To make the paper more accessible, we plan to add more high-level intuition and prose discussions to the main text. Specifically, we will explicitly outline the technical roadblocks, the high-level proof architecture, and the role of our mathematical tools. We address your questions below:
>
> 1. Gap between constants (Question 1): The fundamental obstacle to matching the constants in the numerators of the upper and lower bounds is that, even in the one-dimensional setting ($d=1$), the exact asymptotic behavior of $c_n$ remains unknown. As defined in Equation (5) (line 156, column 1), $c_n$ is the infimum of the $L^1(\phi_1)$-norm of $P$ over all $P \in \Pi_n$ such that the $L^2(\phi_1)$-norm of $P$ is one.
> Currently, we rely on the bounds $n^{-1/4}e^{-n} \lesssim c_n \lesssim 2^{-n/2}$, which leave a gap. On one hand, we derive the lower bound for $c_n$ utilizing the Nikolskii-type and restricted-range inequalities, and we subsequently use $n^{-1/4}e^{-n} \lesssim c_n$ to establish our main theoretical results, including Theorem 2.5. On the other hand, the upper bound $c_n \lesssim 2^{-n/2}$ is witnessed by the sequence of monomials $x^n$ (see line 196, column 2). We leverage this specific sequence to prove our sharpness result for $d=1$, and we then lift this construction to the $d > 1$ setting through the proof of Theorem 4.6 (line 1353). Closing the gap between the constants reduces to determining the exact asymptotic exponent of $c_n$, and it is necessary to go beyond the current techniques of Nikolskii-type and restricted-range inequalities as well as the monomial construction.
>
> 2. Subgaussian tails (Question 2): If we directly apply our methodology to the case of subgaussian mixing distributions, unfortunately, the bound presented in line 148, column 2 does not hold as stated. In the subgaussian setting, the $L^2(\phi_1)$-norm of $r_n$ is $e^{-\Omega(n)}$, not $e^{-\Omega(n\log n)}$ (cf. line 192, column 2), and one might suspect that the sharp inequality between the TV and Hellinger distances would take a different form. It would be an interesting direction for future work to explore the exact relation between TV and Hellinger distances in the case of subgaussian mixing distributions as well as mixing distributions of general tail behaviors.
>
> 3. Condition on $n$ in Theorem 3.1 (Question 3): Please allow us to clarify that the result of Theorem 3.1 indeed holds “for all $n \geq 1$”. This is justified by a relabeling argument as follows. Lemma 3.2 and Corollaries 3.3 and 3.4 hold for all sufficiently large odd numbers $n$. Accordingly, in the proof of Theorem 3.1 (line 321, column 1), we showed that there exists $N_0 \in \mathbb N$ such that $TV(f_{\pi_n^2}, f_{\eta_n^2}) < e^{-e}$ (we replace $\pi^{(2)}$ with $\pi^2$ since the former is not rendered well here) and
> $$
>     H(f_{\pi_n^2}, f_{\eta_n^2}) \geq TV^{1-\alpha^*(TV(f_{\pi_n^2}, f_{\eta_n^2}))}(f_{\pi_n^2}, f_{\eta_n^2})
> $$
> hold for all odd integers $n \geq N_0$. Relabeling indices via the map $n \mapsto 2(n+N_0)+1$, define $\pi_n = \pi_{2(n+N_0)+1}^2$ and $\eta_n = \eta_{2(n+N_0)+1}^2$. Then, these sequences satisfy the statement of Theorem 3.1 for all $n \geq 1$. To assist readers' understanding, we will explicitly add the above explanation to the proof of Theorem 3.1 in the revised manuscript.
>
> 4. Choice of Chebyshev nodes (Question 4): Our choice of Chebyshev nodes is not necessarily because other stable designs would fail badly, but rather because of the immense mathematical advantages and convenience they provide in the proof. This can be summarized into three main points:
>
> First, it is known that the inverse Vandermonde matrix associated with Chebyshev nodes is well-conditioned (Gautschi, 1974; see line 215, column 2).
>
> Second, it enables the control of higher-order moments. As seen in the proof of Lemma 3.2 (lines 1139-1151), to recursively control higher-order moments via lower-order moments using the triangle inequality, the $\ell_1$ norm (sum of absolute values) of the polynomial coefficients is crucial. Statements 1 and 2 of Lemma B.2 show that Chebyshev nodes make this control tractable due to the available $\ell_1$ norm bound in the literature for the coefficients of Chebyshev polynomial.
>
> Third, as given in line 1064, Chebyshev nodes lie in $[-1, 1]$.
> This is convenient in terms of constructing mixing distributions with bounded support.
>
> While these advantages make Chebyshev nodes an excellent choice, we do not claim that they are the only possible design.
>
> 5. Typos (Question 5): We will correct it to “then modifies” as suggested and will conduct a thorough check for additional grammatical errors and typos in the final manuscript.

---

> > ### Author Rebuttal · Reviewer_zMsC · 2026-04-01
> >
> > I think my questions have been adressed. Will keep my score.

---

### Official Review · Reviewer_8nLh · 2026-03-13

**Soundness:** 3
**Presentation:** 2
**Significance:** 3
**Originality:** 4
**Overall Recommendation:** 4
**Confidence:** 3

**Summary:**

This paper studies the relation between total variation and Hellinger distances for Gaussian location mixtures with compactly supported mixing distributions. The main result shows that the Hellinger distance can be controlled by a $1-o(1)$ power of the total variation distance, and the paper also provides a matching sharpness construction showing that the exponent cannot, in general, be improved to $1$. The paper further derives consequences for learning Gaussian mixtures in total variation, robust density estimation under Huber contamination, and empirical Bayes regret bounds.

**Compliance With Llm Reviewing Policy:**

Affirmed.

**Final Justification:**

My concerns have been addressed. I remain positive on the paper.

**Key Questions For Authors:**

1. Can the authors state more explicitly which parts of the robust estimation result are minimax-optimal and which parts remain open?
2. Can the authors move a quantitative summary of the $M,d$-dependence of $C_0$ from Appendix A.3 into the main text?
3. Can the authors provide a fuller proof of the subadditivity step for $J(t)$ and expand the derivation of Theorem 4.5 in Appendix C?

**Limitations:**

The paper does not yet adequately discuss limitations and potential negative societal impact. The authors should explicitly discuss the restriction to compactly supported isotropic mixtures, the potentially strong hidden dependence on $M$ and $d$, the fact that only the $\epsilon$-dependence is shown to be sharp in the robust estimation application, and the mainly theoretical rather than practical nature of the Yatracos-based estimator and regret bound. The impact statement should also address the risk of overstating “robust” guarantees in contaminated or high-stakes settings.

**Strengths And Weaknesses:**

## Strengths

1. The main theorem addresses a natural open question and is strengthened by a nontrivial sharpness result, which makes the contribution more than just an upper bound.
2. The technical development appears substantial. The Hermite expansion, together with the Nikolskii-type and restricted-range inequalities, is well matched to the problem.
3. The core technical contribution is likely of independent interest beyond the specific applications presented in Section 4.

## Weaknesses

1. The robustness claims are somewhat overstated. The abstract and discussion suggest “optimal robust estimation,” but Theorem 4.5 gives an upper bound with both an $\epsilon$-term and an $n^{-(1-o_d(1))}$ term, while Theorem 4.6 only matches the $\epsilon$-dependence. The paper also explicitly notes that the sample-size term is likely improvable. The presentation should therefore distinguish more clearly between what is sharp and what is not.
2. The dependence on support radius and dimension is hidden too aggressively. While the exponent is presented as independent of $M$ and $d$, Appendix A.3 shows that $\log C_0$ has polynomial order in $M^2 d$. Since Section 4 suppresses these constants, the resulting rates look cleaner than they actually are. This dependence should be stated much more prominently in the main text.
3. The proof of Theorem 4.5 is too compressed in its current form. In particular, the argument relies on subadditivity of $J(t)=C_0 t \vee t^{1-\alpha(t)}$, but this is only asserted. The step from the TV control in Proposition 4.4 to the expectation bound in Theorem 4.5 also reads more like a proof sketch than a complete derivation. This part should be expanded.
4. The Section 4.2 applications are mainly theoretical, but the exposition does not state this clearly enough. The estimator is based on the Yatracos construction, and the regret bound in Theorem 4.7 uses a tuning parameter $\rho(\epsilon,n)$. That is acceptable for a minimax upper bound, but the paper should be more explicit that this is a theoretical consequence of the inequality rather than a practically implementable robust empirical Bayes procedure.

---

> ### Author Rebuttal · Authors · 2026-03-31
>
> We sincerely thank the reviewer for recognizing our work and for providing constructive and insightful feedback. We address your comments and questions below, and we will incorporate all of your suggestions into the revised manuscript to improve clarity and transparency.
>
> 1. Minimax Optimality and Robustness Claims (Weakness 1 \& Question 1): We fully agree that the presentation should distinguish more clearly between what is sharp and what is not. The upper bound we present in Theorem 4.5 is minimax optimal with respect to the $\epsilon$-dependence. As mentioned at the end of the Introduction (line 068, column 1), this establishes the minimax optimal rate in the large-$\epsilon$ regime where $n \geq \text{poly}(1/\epsilon)$ and the $\epsilon$-term dominates the $n$-term. This is intended to showcase a direct application of our theoretical results. Furthermore, it is worth noting that determining the optimal dependence on $n$ is a long-standing open question, also raised as an open question in Jia et al. (2023), even in the setting without contamination ($\epsilon = 0$). To prevent any overstatement, we will explicitly clarify in the revised manuscript that while our $\epsilon$-dependence is sharp, the $n$-dependence remains an open problem and is likely improvable.
>
> 2. Dependence of $C_0$ on $M$ and $d$ (Weakness 2 \& Question 2): The dependence of the constant $C_0$ on the support radius $M$ and dimension $d$ is derived in Appendix A.3. We placed these details in the appendix due to space constraints. In the revised manuscript, we will ensure full transparency by moving a quantitative summary of the dependence from Appendix A.3 directly into the main text. We thank you again for the constructive suggestion.
>
> 3. Proof of Theorem 4.5 and Subadditivity of $J(t)$ (Weakness 3 \& Question 3): We acknowledge that our proof of Theorem 4.5 is currently compressed. Accordingly, we will provide an expanded and detailed proof in the revision as follows: We first show that the function $G(t) := t^{1-\alpha(t)}$ is strictly increasing and concave on an open interval $t < t_0$, where $\alpha(t) = \frac{c}{\log\log(1/t)}$, $c = 2+\delta$, and $t_0 < e^{-e}$ is a constant depending only on $\delta$. To this end, we take derivatives:
> $$
>     G'(t) = \frac{G(t)}{t}(1-\frac{c}{\log\log(1/t)}+\frac{c}{\log^2\log(1/t)}),
> $$
> $$
>     G''(t) = -\frac{G(t)}{t^2}(\frac{c}{\log\log(1/t)}-\frac{c^2+c+o(1)}{\log^2\log(1/t)}),
> $$
> verifying that $G'(t) > 0$ and $G''(t) < 0$ hold for all $t < t_0$. We also note that $\lim_{t \searrow 0} G(t) = 0$. Therefore,
> given that $C_0$ is not less than $t_0^{-\alpha(t_0)}$, a constant depending only on $\delta$, there exists $0 < t_1 \leq t_0$ such that
> $$
>     J(t) = C_0 t \vee G(t) = \begin{cases}
>         C_0 t, & t \geq t_1, \\\\
>         G(t), & t < t_1.
>     \end{cases}
> $$
> Using the concavity of $G$, we obtain for all $s,t>0$ that
> $$
> \begin{aligned}
>     J(s+t) &= C_0(s+t) \vee G(s+t)
>     \\\\ &\leq C_0(s+t) \vee (G(s)+G(t))
>     \\\\ &\leq (C_0s \vee G(s)) + (C_0t \vee G(t))
>     \\\\ &= J(s) + J(t).
> \end{aligned}
> $$
> The second line is due to the fact that $C_0(s+t) < G(s+t)$ implies $G(s+t) \leq G(s)+G(t)$, and the third line is due to the general inequality: $(a+c)\vee(b+d)\leq(a\vee b)+(c\vee d)$.
> This justifies line 1348. Hence, by Cauchy-Schwarz inequality,
> \begin{align*}
>     H^2(f_\pi, \widehat f) \leq (3^2+3^2+2^2)[J^2(\epsilon) + J^2(\eta) + J^2(\mathrm{dist}(P, \widehat P_n))].
> \end{align*}
> Taking expectations on both sides yields the desired bound by choosing the same $\eta$ as in the proof of Proposition 4.4.
>
> 4. Theoretical Nature of Applications and Limitations (Weakness 4 \& Limitations): We agree that the exposition should more clearly state the purely theoretical nature of the robust estimation results. The Yatracos-based estimator is a theoretical construction for the total variation loss, and it is not computable in practice. On the other hand, the dependence of the tuning parameter $\rho(n,\epsilon)$ on $\epsilon$ can be alleviated by the standard Lepski's method to achieve adaptive estimation when $\epsilon$ is unknown. Whether there exists a polynomial-time algorithm achieving the same guarantee remains an interesting open problem. While it is possible to show that a kernel density estimator achieves a near-optimal rate in $\epsilon$, we are not aware of a polynomial-time algorithm that attains the exact optimal rate. In the revision, we will discuss the theoretical nature of the results more explicitly in Section 4.2.
>
> Furthermore, we will expand our impact statement to explicitly address the following points: (1) The restriction of our results to compactly supported isotropic mixtures; (2) The fact that only the $\epsilon$-dependence is shown to be sharp in our robust estimation bounds, while the $n$-dependence remains open; (3) The purely theoretical nature of the Yatracos-based estimator and regret bounds. We believe this will address the risk of any overstatement.

---

> > ### Author Rebuttal · Reviewer_8nLh · 2026-04-03
> >
> > Thank you for the detailed rebuttal. I remain positive on the paper and keep my score.

---

### Official Review · Reviewer_oahL · 2026-03-13

**Soundness:** 3
**Presentation:** 2
**Significance:** 3
**Originality:** 3
**Overall Recommendation:** 4
**Confidence:** 3

**Summary:**

This paper considers the following family of distributions called Gaussian mixtures: $\{ f_\pi \}$ where $f_\pi$ is the convolution of a distribution over $\mathbb{R}^d$ $\pi$ and the standard Gaussian. Without any assumptions on two distributions $p, q$ we know that the following two inequalities hold between Hellinger distance and the TV distance: $H^2(p, q) \le TV(p, q) \le \sqrt{2} H(p, q)$. The focus of this work is studying this relationship between TV and Hellinger when $p, q$ are guaranteed to be in the form of $f_\pi$ when $\pi$ is guaranteed to be supported over some ball of radius constant. Earlier work of (Jia et al 2023) showed that for the case of KL and Hellinger we have that they are asymptotically the same for Gaussian mixtures and used that to get entropy-based characterizations of Hellinger/KL estimation risk, but for TV they only had the general bound of $H^2 \le TV \le H$ and asked if one can upgrade the lower bound to just $H$, when $\pi$ is supported on a bounded domain. This work shows that $H \le TV^{1 - o(1)}$ for Gaussian mixtures of distributions with bounded support, and shows that the $o(1)$ term is actually necessary and so this bound is actually tight.

As an application, they show that for density estimation of such Gaussian mixtures under Huber contamination, the rate is $\epsilon^{1 - o(1)}$ provided $n = \text{poly}(1/\epsilon)$ samples are taken.

**Compliance With Llm Reviewing Policy:**

Affirmed.

**Final Justification:**

I am keeping my score at 4, but I am strongly positive on the paper and think it should be accepted. My hesitation about moving to 5 is about impact, not quality. The problem feels somewhat specialized to me, so I am unsure how broadly influential the result will be. I would defer to reviewers with deeper expertise within this subarea on whether the techniques are strong enough to merit a stronger outcome.

**Key Questions For Authors:**

No major questions. I recommend adding a discussion of the proof techniques and comparing them against prior work.

**Limitations:**

yes

**Strengths And Weaknesses:**

I have not checked the proofs too carefully but they seem sound. The presentation is okay. I think a discussion of the techniques in prose would be helpful, the way it's currently written it's not clear where the main technical hardness is. I think a paragraph explaining why the previous work / techniques failed at this or essentially answering "what has changed in the proof architecture and analysis? what were the roadblocks?" would be very helpful. This paper studies a fundamental and simple setting and answers an open question from prior work (Jia et al 2023), and surprisingly refutes it but still shows a bound very close to the conjectured bound. The application to robustly learning Gaussian mixtures might be interesting to the robust stats community, though I think studying learning under Hellinger distance is less studied. The proofs seem technical and novel.

Overall I lean towards accepting this paper.

---

> ### Author Rebuttal · Authors · 2026-03-31
>
> We sincerely thank the reviewer for the positive evaluation and constructive feedback. We agree that adding a prose discussion of the proof techniques would improve the clarity of the paper. In the revised manuscript, we will add a paragraph explaining our main technical tools and how they differ from prior work, as detailed below:
>
> 1. Differences from Prior Work (Jia et al.): In Jia et al. (2023), the main results rely on a direct analysis of the ratio of Gaussian mixture densities to establish the equivalence between the KL divergence and the squared Hellinger distance. While it is unclear whether their approach can be directly adapted to our setting—which compares the Total Variation (TV) and Hellinger distances—we instead take a different, moment/polynomial-based approach. As shown in Section 2, comparing the TV and Hellinger distances essentially reduces to establishing an inequality between the $L^1(\phi_d)$ and $L^2(\phi_d)$ norms of polynomials (cf. Theorem 2.5 and its proof). Consequently, Hermite polynomial expansions combined with precise moment control provide a more natural and suitable approach.
>
> 2. Main Technical Tools: We summarize our technical tools for both upper and lower bounds below:
>
> Upper bound: Establishing this bound requires tight control over the moments in multi-dimensional settings. To achieve this, we introduce a multi-dimensional generalization of Nikolskii-type (Proposition A.6) and restricted-range (Proposition A.7) inequalities, which were previously well-understood in 1D.
>
> Lower bound (Sharpness): Demonstrating the sharpness of our bound requires us to explicitly construct a sequence of mixing distributions. This construction necessitates an analysis of Chebyshev nodes. For further details on our choice of Chebyshev nodes, please refer to our response to Question 4 from Reviewer zMsC.

---

> > ### Author Rebuttal · Reviewer_oahL · 2026-04-04
> >
> > Thank you for the rebuttal. I appreciate the clarification on the main technical ideas and how your approach differs from prior work. I remain positive overall and am keeping my score unchanged. That said, my concern was not just identifying the main tools, but having the paper guide the reader through the overall proof strategy. At present, the presentation still feels too theorem-driven, with not enough prose explaining how the pieces fit together and why this is the right approach. So while the rebuttal helps clarify the headlines of the techniques, I still think the paper would benefit from more intuition and narrative guidance in the main text.

---

### Decision · Program_Chairs · 2026-04-30

**Decision:**

Accept (spotlight)

**Comment:**

This is a strong theoretical paper. Reviewers agreed that the paper resolves a natural open question about the relationship between total variation and Hellinger distances for bounded-support Gaussian mixtures. In particular, the Hellinger distance is shown to be upper bounded by TV to a power 1-o(1), and in addition, the o(1) term is show to be necessary. The proofs seem to have substantial depth. The paper shows how this  result can be used in specific applications (e.g. density estimation of such Gaussian mixtures under Huber robustness), and it generally seems like a result that would be of broad interest.

I therefore recommend acceptance. Since the paper is highly technical, the final version could benefit from more explanations around the proof strategy.